# Mexico's High Resolution Climate Database (MexHiResClimDB): a new daily high-resolution gridded climate dataset for Mexico covering 1951–2020

Jaime J. Carrera-Hernández[1]

[1]Instituto de Geociencias, Universidad Nacional Autónoma de México (UNAM), Querétaro, México

**Correspondence:** J. J. Carrera-Hernández (jaime-carrera@geociencias.unam.mx)

**Abstract.** This work presents Mexico's High Resolution Climate Database (MexHiResClimDB), which is a newly developed gridded, high-resolution climate dataset comprised of daily, monthly and yearly precipitation and temperature ($T_{min}$, $T_{max}$, $T_{avg}$). This new database provides the largest temporal coverage of the aforementioned climate variables at the highest spatial resolution (20 arc sec, or 560 m on Mexico's CCL projection) when compared to the other currently available gridded datasets for Mexico and its development has allowed for the analysis of the country's climate extremes for the 1951–2020 period. By comparing the spatial distribution of precipitation from the MexHiResClimDB with other gridded data (Daymet, L15, CHIRPS and PERSIANN CDR), it was found that the precipitation provided by this new dataset adequately represents the spatial variation of extreme precipitation events, in particular for the precipitation that occurred during September 15–16 of 2013, caused by the presence of Tropical storm Manuel in the Pacific Ocean and Hurricane Ingrid (Cat 1) in the Gulf of Mexico. Using data from 61 days retrieved from Automated Weather Stations located throughout Mexico – and correspoding to the two months with the largest precipitation in Mexico – it was found that precipitation data from MexHiResClimDB has the lowest MAE (8.7 mm), compared to those of L15 (9.5 mm), Daymet (10.1 mm) and CHIRPS (11.7 mm). For $T_{min}$ and $T_{max}$, the lowest MAE was obtained with MexHiResClimDB (1.7°C and 1.8 °C, respectively), followed by Daymet (2.0 °C for both temperatures) and L15 (2.4°C and 2.5 °C). With this new database an analysis of the extreme events of precipitation and temperature in Mexico for the 1951–2020 period was undertaken: the wettest year was 1958, the wettest day 1970-09-26, and September of 2013 the wettest month. It was also found that eight out of the ten days with the highest $T_{min}$ occurred in 2020, the two months with the highest $T_{min}$ were July and August of 2020 and that the six years with the highest $T_{min}$ were 2015–2020. When $T_{max}$ was analysed, it was found that the hottest day was 1998-06-15, while June of 1998 was the hottest month and 2020 the hottest year, and that the four hottest years occurred between 2011–2020. Nationwide (and considering 1961–1990 as the baseline period), $T_{min}$, $T_{avg}$ and $T_{max}$ have increased, with their anomalies drastically increasing in recent years and reaching values above 1.0 °C in 2020. At the same time, precipitation has also decreased in recent years – which combined with the increase in temperature will have severe impacts on water availability. This new database provides a tool to quantify – in detail – the spatio-temporal variability of climate throughout Mexico.

The MexHiResClimDB entire dataset is available on Figshare (DOI:10.6084/m9.figshare.c.7689428, Carrera-Hernández (2025a))

## 1 Introduction

Gridded climate data are important because regional changes are highly spatially heterogeneous (Walther et al., 2002), and long-term climate information is of primary importance to estimate groundwater recharge (Carrera-Hernández and Gaskin, 2008a; Carrera-Hernández et al., 2012, 2016), to study floods, droughts (Wehner et al., 2011), heatwaves or changes in the water cycle intensity (Huntington et al., 2018). Furthermore, the spatial distribution of climate variables is required not only for the development of water management related analyses, but also to study the distribution of vegetation (Sáenz-Romero et al., 2010), shifts in the composition of plant communities (Feeley et al., 2020), to identify potential areas for resting, feeding and reproduction along migratory routes of butterflies (Castañeda et al., 2019), to develop niche-based species distribution models (Perez-Navarro et al., 2021), to quantify the main drivers of extinction risk of terrestrial animals and vascular plants (Esperon-Rodriguez et al., 2024), or to locate conservation hotspots for reptiles (Ramírez-Arce et al., 2024).

Due to the importance of climate and of quantifying its spatial distribution through time, several authors have developed gridded datasets – with different geographic and temporal coverages – of various climate variables: Hijmans et al. (2005) developed the WorldClim dataset, which comprises interpolated monthly climate surfaces ($T_{min}$, $T_{avg}$, $T_{max}$ and precipitation) for global land areas covering the 1950–2000 period at a resolution of 30' ($\approx$1 km at the equator). This dataset was updated by Fick and Hijmans (2017) who created a monthly dataset for the 1970–2000 period using the thin-plate smoothing algorithm implemented in ANUSPLIN (Hutchinson, 2007) using covariates such as elevation, distance to the coast and three satellite-derived covariates (maximum and minimum land surface temperature as well as cloud cover obtained from MODIS). Becker et al. (2013) document the global land-surface precipitation data products of the monthly Global Precipitation Climatology Centre (GPCC at a $0.25°$ – or 15' – resolution, $\approx$25 km), which was later improved by Schneider et al. (2014). Other datasets provide gridded data not only for precipitation, but for other climate variables as well, such as Terraclimate (Abatzoglou et al., 2018), which is a world-wide monthly climate dataset (Precip, $T_{min}$, $T_{max}$, wind speed, vapor pressure and solar radiation) for the 1958–2024 period at a resolution of $\approx$4 km, which was developed using the climate normals from the WorldClim dataset along with monthly data from other sources.

In order to improve the temporal and/or spatial resolution of globally available datasets, several authors have developed gridded climate datasets for different countries: for Croatia, Tadić (2010) developed maps for the 1961–1990 normals of 20 climate variables using 567 weather stations and a spatial resolution of 1 km, while Yatagai et al. (2012) developed a daily gridded precipitation dataset for Asia (APHRODITE) at a resolution of $0.25°$ ($\approx$25 km). For Finland, Aalto et al. (2016) developed FMI ClimGrid for the 1961–2010 period with a resolution of $10\times10$ km$^2$ for an area of $\approx$338,000 km$^2$; Hollis et al. (2019) developed HadUK-Grid, which is a dataset of interpolated observations of daily temperature (maximum, mean and minimum starting in 1960), precipitation (since 1891) and other monthly variables at a resolution of $1\times1$ km$^2$. More recently, Razafimaharo et al. (2020) updated the HYRAS dataset (mean, minimum and maximum temperature along with relative humidity) developed in 2014 for Germany (357,596 km$^2$) at a resolution of $5\times5$ km$^2$ for the 1951–2015 period through the use of Inverse Distance Weights (IDW, Frick et al. (2014)); for Serbia, Sekulić et al. (2021) developed MeteoSerbia1km, which is a daily gridded meteorological dataset at a 1 km$^2$, covering an area of 88,361 km$^2$, while Xavier et al. (2022) developed

daily weather gridded data for 1961–2020 for Brazil using Inverse Distance Weights (IDW). For some countries – like Spain – there are different gridded datasets: the Spanish PREcipitation At Daily scale (SPREAD) dataset (Serrano-Notivoli et al., 2017), developed for the 1950–2012 period at a 5×5 km resolution for peninsular Spain (with an area of 494,011 km$^2$), the Monthly Precipitation dataset (MOPREDAScentury, Beguería et al. (2023)), developed for the 1916–2020 period, with a spatial resolution of 0.1° (≈10 km), or the Iberia01 (Herrera et al., 2019), which comprises daily gridded data for the 1971–2015 period with the same spatial resolution (≈10 km). For a more in-depth review of gridded climate products, interested readers are referred to the recent work of Mankin et al. (2025), who reviewed a total of 63 gridded climate datasets.

For North America (Mexico, United States and Canada) the currently available gridded climate datasets that incorporate precipitation and temperature are the L15 (Livneh et al., 2015) with a resolution of approximately 6 km, and Daymet (Thornton et al., 2021), with a resolution of 1 km; the temporal coverage of the previously mentioned datasets is 1951–2015 for L15, while daymet starts in 1980 and is updated yearly. For the contiguous United States (CONUS), there are several gridded climate datasets, with the Parameter-elevation Regressions on Independent Slopes Model (PRISM, Daly et al. (1997)) being probably the first to use auxiliary datasets to estimate precipitation and temperature, while nClimGrid-Daily (Durre et al., 2022) is the most recent, providing temperature and precipitation at a resolution of ≈5 km for the 1951–2022 period. For Canada, the Natural Resources Canada observational dataset (NRCANmet, Hopkinson et al. (2011)) is availabe at a resolution of 10 km and provides daily data of precipitation along with minimum and maximum temperature for the 1950–2008 period. For Mexico, Englehart and Douglas (2004) developed monthly maps of surface air temperature at a resolution of 2.5° × 2.5° for the 1940–2001 period using data from 103 stations. Sáenz-Romero et al. (2010) developed interpolated surfaces of $T_{min}$, $T_{avg}$ and $T_{max}$ and precipitation for monthly normals of the 1961–1990 period at a spatial resolution of 1×1 km. Fernández-Eguiarte et al. (2012) developed the Climatological Atlas for Mexico, which consists of monthly gridded data of Precipitation, $T_{min}$, $T_{avg}$ and $T_{max}$ averaged for the 1903–2010 period (i.e. 12 monthly rasters per climate variable, for a total of 48 rasters that cover the entire period) at a spatial resolution of 926 metres. This database was later updated in order to provide monthly data of the aforementioned variables for years 1979–2009. Cuervo-Robayo et al. (2014) developed monthly surfaces of precipitation, $T_{min}$ and $T_{max}$ for 1910–2009 (i.e. 12 surfaces in total) at a 30 arc-sec resolution (≈900 m), and more recently (Cuervo-Robayo et al., 2020) averaged data for three periods (1910–1949, 1950–1979, 1980–209), using the same methodology and spatial resolution as in their previously mentioned study (interpolation with ANUSPLIN and 900 m$^2$). From this summary, it can be seen than to date, there is no daily high resolution climate dataset available for Mexico, which is why the Mexico's High Resolution Climate Database (MexHiresClimDB), which covers the 1951–2020 period at a spatial resolution of 20" (≈600) meters was developed, as described in this work.

## 2  Study area

Mexico – with a continental area of $1.96×10^6$ km$^2$ – is bordered to the west by the Pacific Ocean and to the East by the Gulf of Mexico, with an abrupt topography that varies from sea level up to nearly 5,700 m a.s.l. (Fig. 1(a)). Due to the interaction between warm and cold air masses and Mexico's topography, climates that range from very arid to humid are found within it,

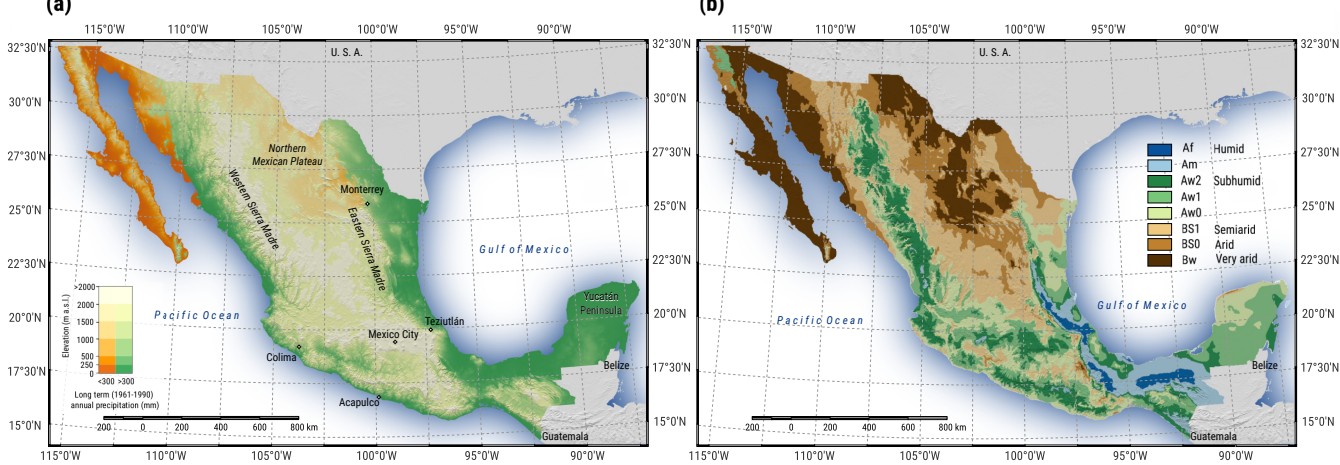

**Figure 1.** Mexico's (a) topography and (b) climates. The cities shown – except Mexico City, which is only shown as reference – have had catastrophic floods and landslides caused by to different Tropical Cyclones. Mexico is shown in Lambert Conformal Conical projection with shaded relief derived from the AW3D30 DSM and a cross-blended hypsometric color scale (Patterson and Jenny, 2011) to differentiate between arid and non-arid regions (the color scale varies according to the precipitation normal of 1961–1990); the distribution of climates in Mexico is adapted from García (2004)

with vegetation that varies from xerophyle shrubs and grasslands to tropical forests and even cloud forests (Fig. 1(b)). Mexico's main sierras – the Western and Eastern Sierras, which run nearly parallel to the Pacific and Gulf coasts, as shown in Fig. 1(a) 95 – block moist air masses from the aforementioned coasts. Accordingly, the area enclosed by Mexico's Sierras – the Northern Mexican Plateau – has semiarid to very arid climates, while humid climates are found on the windward side of these Sierras (Fig. 1(b)). Furthermore, Tropical Cyclones that originate from both the Atlantic and the Eastern Pacific basin make landfall in Mexico (Farfán et al., 2014); in fact, in the second half of the 20th century, a total of 65 hurricanes impacted Mexico's Pacific coast, while 27 impacted its eastern coast (Jauregui, 2003). Due to Mexico's geographic context, its precipitation is extremely 100 variable, as on a given day during the rainy season, precipitation can vary from 0 to more than 300 mm and the adequate representation of these extreme events is important in Mexico, in particular for cities that depend on reservoirs that are filled by hurricanes and tropical storms or where flash-floods are a recurrent event.

## 3   Methodology

The climate data used in the present study was downloaded from Mexico's meteorological service website (last date consulted: 105 December 2024) on a station by station basis through a `bash` script, and further processed with a series of both `awk` and `R` scripts in order to generate a `PostgreSQL` relational database, as proposed by Carrera-Hernández and Gaskin (2008b). This database can be accessed through both `R` and the Open Source GRASS GIS, which allows the use of data stored in GRASS

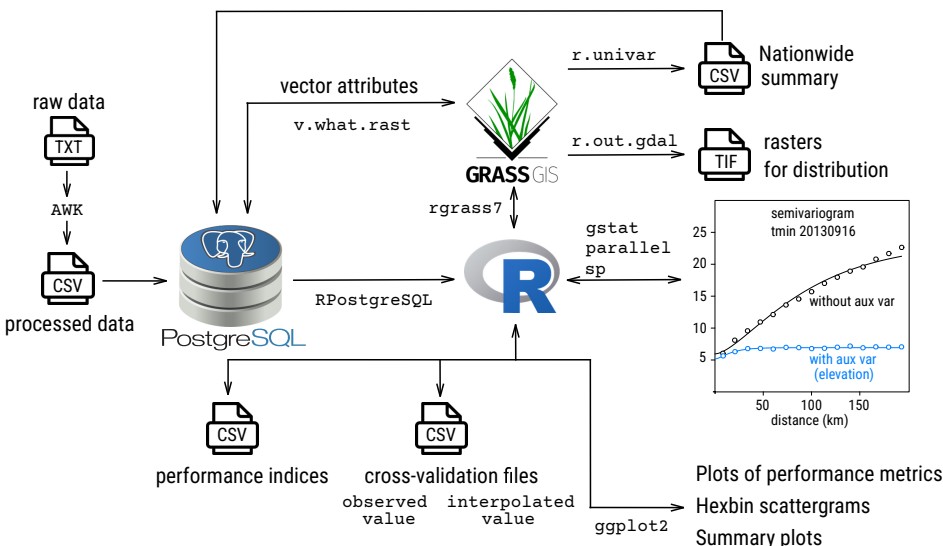

**Figure 2.** Flowchart describing how the relational database developed in this work can be accessed with R and the Open Source GRASS GIS. This flowchart also shows that cross-validation files were created for each variable on a daily basis in order to compute different metrics to evaluate the performance of the interpolation method applied in this work. The effect of considering elevation as an auxiliary variable to apply Kriging with External Drift is also shown on the $T_{min}$ semivariogram for September 16th of 2013.

to be used by different R libraries and to store the spatial data created in R in GRASS's internal file structure, as detailed in Figure 2.

To develop the relational database, the location of all weather stations was first verified (as some stations had wrong coordinates) and once the climate records were in PostgreSQL only those stations with more than 80% of registered data for at least 10 continuous years were selected (wheras the L15 dataset (Livneh et al., 2015) used a 50-day threshold for stations in Mexico). Accordingly, the number of stations used to undertake the interpolations across the 1951–2021 period varies, as shown in Fig. 3. This Figure shows that there are fewer records of temperature than precipitation (a situation also observed

for North America by Tang et al. (2020)) and that the maximum number of records for both variables were registered in years 1982 and 1983 (with over 4000 daily records for precipitation and between 3000–3500 for temperature out of a total of 5467 stations from the downloaded raw data), and that years 1951 and 2021 exhibit the lowest number of records. However, it should be kept in mind that a larger number of records does not represent better spatial coverage, as shown in Fig. 4, where it can be seen that by 2021 the spatial coverage of weather stations is limited outside Mexico's central region – which is why the

MexHiResClimDB does not include data after year 2020. A basic Quality Control (QC) was applied to the generated database, with daily precipitation values above 600 mm, as well as temperature values below -30 °C or above 60 °C being discarded, along with days where $T_{min} > T_{max}$.

Although the removal of outliers considering neighbouring stations could have been done, this was not done due to the fact that precipitation in Mexico is highly variable within short distances due to the presence of hurricanes and these precipitation

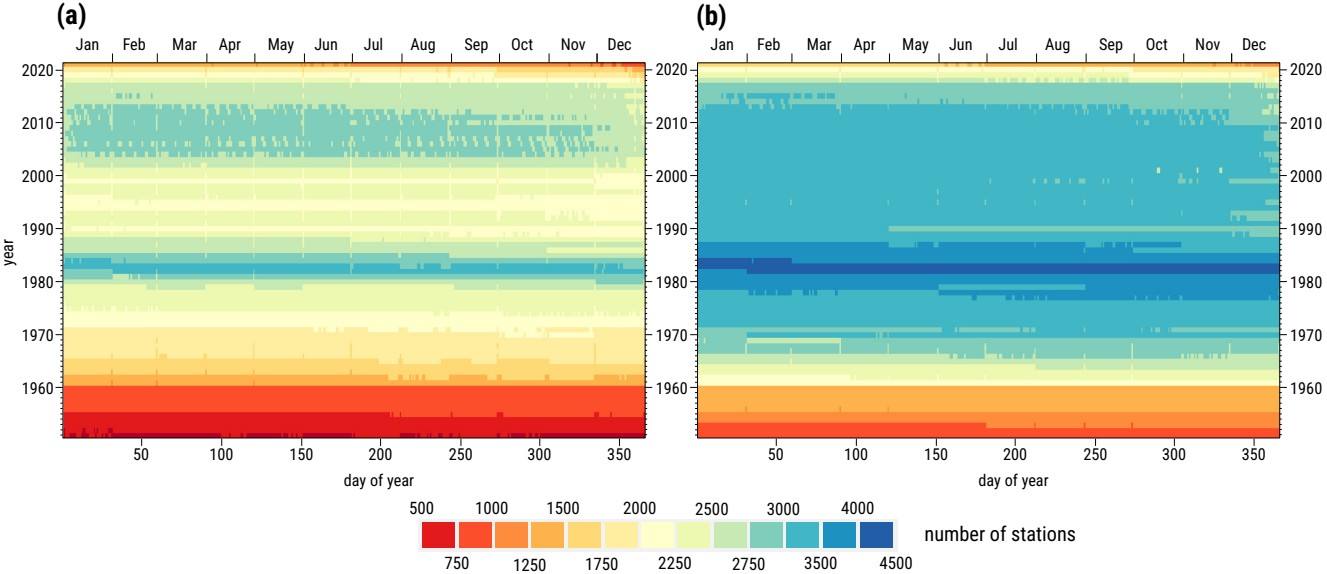

**Figure 3.** Number of weather stations used for daily interpolations of: (a) Temperature, (b) Precipitation. It can be seen that 1982 and 1983 were the years with the largest number of stations and that a decrease of records for both temperature and precipitation started in 2013.

events need to be included in the gridded product. In addition, the final data selected by the previously mentioned procedure were not analysed for homogeneity and the station records were used without filling data gaps (i.e. data series reconstruction). Although gap-filling can be used to generate a complete data series for the considered time period – and thus keeping a uniform number of stations for the interpolation – the decision was made not to do it in order to only use the original data. The reconstruction process is generally based on weighted averages or modeling that consists of creating a reference series formed as a weighting model of the data observed at neighbouring stations (Serrano-Notivoli and Tejedor, 2021), which adds data using a method of interpolation (Daly, 2006). Further work can be done to address these issues and interested readers are referred to Serrano-Notivoli and Tejedor (2021) for a detailed analysis of Quality Control on the development of gridded climate datasets.

The interpolations were developed using Kriging with External Drift on a local neighbourhood (KED$_l$) using topography as an auxiliary variable because it has been found to be an adequate technique to interpolate both precipitation and temperature on a daily and yearly basis (Carrera-Hernández and Gaskin, 2007; Page et al., 2022; Carrera-Hernández et al., 2025). The ALOS AW3D30 DEM was used as auxiliary variable because it has been shown to better represent Mexico's topography (Carrera-Hernández, 2021), and the original 1 arc-sec AW3D30 DEM was resampled to a 20 arc-sec resolution using the `r.neighbors` command of the GRASS GIS. There are currently several books and articles that describe the equations and assumptions behind Kriging and its different variants (Isaaks and Srivastava, 1989; Goovaerts, 1997, 2000; Carrera-Hernández and Gaskin, 2007; Pebesma, 2014); accordingly, they are not discussed in this work. Interested readers can find examples of the implementation of Kriging with `R` and `gstat` in Pebesma and Benedikt (2023) and Pebesma and Bivand (2023).

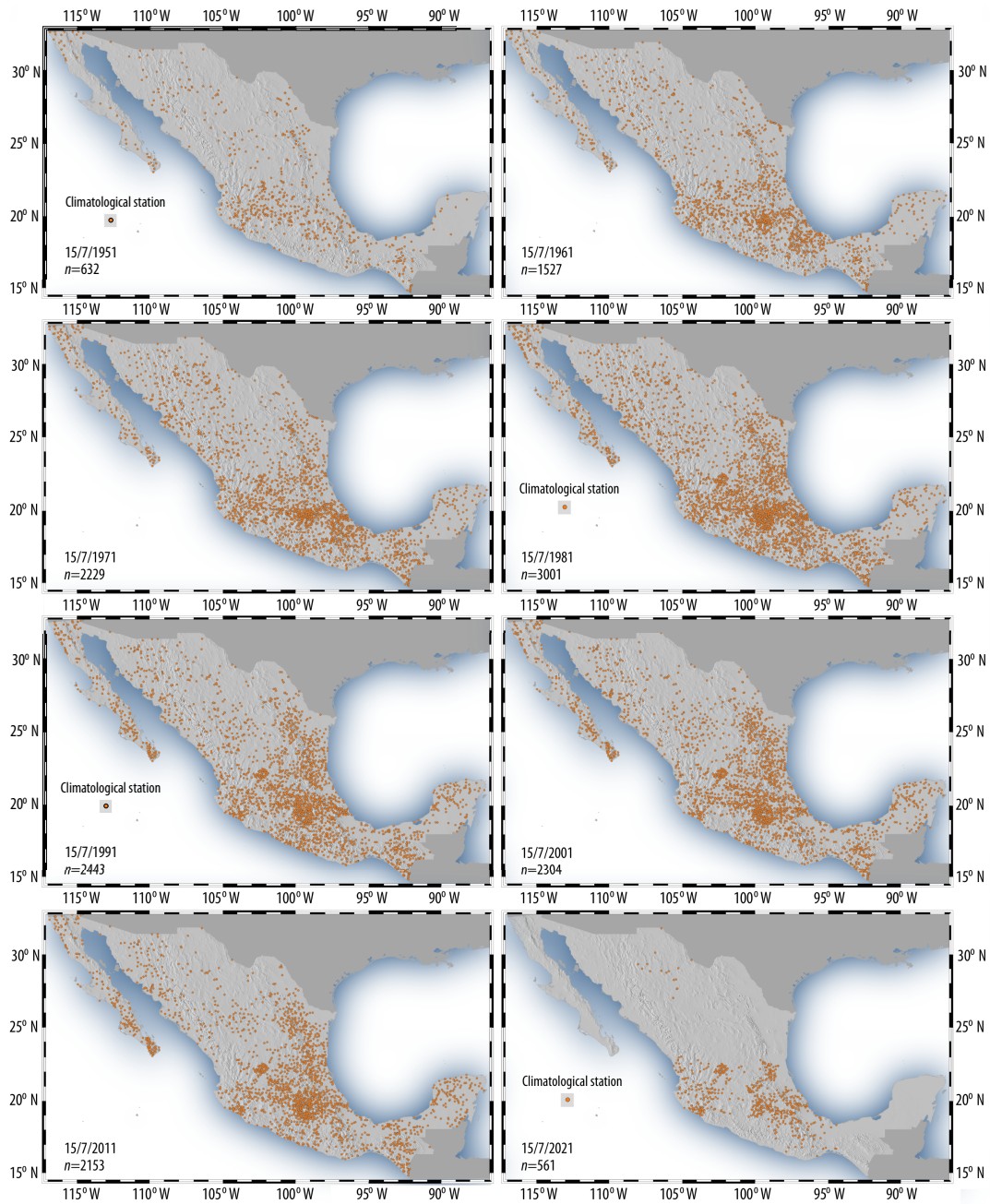

**Figure 4.** Spatial coverage of weather stations from 1951–2021. It can be seen that in 2021 the spatial coverage is mainly limited to Mexico's central region, which is why even though there are climate data for 2021, the MexHiResClimDB does not include 2021.

The interpolations were carried out using the GRASS GIS and R with the libraries `RPostgreSQL`, `parallel`, `gstat`, `sp` and `rgrass7` (Conway et al., 2023; Pebesma and Bivand, 2023). The climate records (i.e. time series data) were stored in PostgreSQL while all the generated raster files were stored in the file structure of GRASS (Fig. 2). A Gaussian semivariogram was fitted to the experimental semivariogram of each variable for each day using GSTAT's automated fitting procedure – which uses an iterative reweighted least squared estimation (Pebesma, 2014). However, some semivariograms had to be manually fitted when the automated procedure could not fit the semivariogram parameters. Based on the author's previous work on nation-wide interpolation of yearly precipitation in Mexico, a cut-off distance of 180 km and a local neighbourhood of 30 stations was used, because $KED_l$ with these parameters adequately interpolates precipitation even when this process is anisotropic (Carrera-Hernández and Gaskin, 2007; Carrera-Hernández et al., 2025). These parameters were recommended by Carrera-Hernández et al. (2025) after a detailed comparison of different Kriging variants at the national level using both global and stratified domains that used different auxiliary variables with a combination of different cut-off distances and local neighbourhoods. This comparison showed that $KED_l$ using elevation as a secondary variable with a cut-off distance of 180 km and a local neighbourhood of 30 stations provided the best representation of yearly precipitation in Mexico (a code snippet is provided in Carrera-Hernández (2025p)).

Each daily interpolation took 26 GB of RAM and a processing time of approximately 16 minutes; and they were carried out on three workstations with multi-core processors and 256 GB of RAM located at the Hydrogeomatics Laboratory of the Geosciences Institute, UNAM. As previously mentioned, the interpolations were carried out using R's `parallel` library and it was found that the use of five cores for each interpolation provided the fastest interpolation time ($\approx$16 minutes, because parallelizing the process also requires time). For each variable 25,564 rasters were interpolated, thus yielding a total of 76,692 rasters, which required a computation time of 28 months (although this time was larger because some days required a manual adjustment of the semivariograms).

In order to estimate the errors in spatial climate datasets, a combination of approaches should be used, involving data that are as independent from those used to generate the datasets, along with common sense in the interpretation of results (Daly, 2006). Accordingly, the MexHiResClimDB was validated using : 1) leave-one-out cross validation – from which different performance metrics were obtained, 2) visual comparison of extreme precipitation events, 3) use of independent data acquired at different Automatic Weather Stations (AWSs) located throughout Mexico, and 4) comparison of the spatial distribution of accumulated monthly precipitation at both national and watershed scales.

## 4  Validation

The first validation step consisted of leave-one-out cross validation (which computes an interpolated value at the location of each station used, without using its value for interpolation) of each variable for each day, which were written as `.csv` files for further analyses. Using these files, four different measures of error were used to validate the interpolations: Coefficient of Determination ($R^2$), Coefficient of Efficiency (COE), Mean Absolute Error (MAE), and Index of Agreement (IOA), which are explained in detail by Legates and McCabe (1999). The COE, MAE and IOA are reported in this work because average-error

and agreement measures based on sums of error magnitudes are – in general – superior to comparable measures based on sums of squared errors (i.e. RMSE, Willmott et al. (2015)). Furthermore, the RMSE is not reported in this work because it is a function of the MAE, the variablity within the distribution of errors and the area represented by the interpolated variable (Willmott and Matsuura, 2006).

In addition to the aforementioned indices, the coefficient of determination ($R^2$) is also reported in this work due to the ease of its interpretation, as it describes the proportion of the total variance in the observed data that can be explained by the model (i.e. if $R^2$=0.80, then the model explains 80% of the variability in the observed data) and is given by:

$$R^2 = \left( \frac{\sum_{i=1}^{n}(Tm_i - \overline{Tm})(T_i - \overline{T})}{\sum_{i=1}^{n}\left(Tm_i - \overline{Tm}\right)^{0.5}\sum_{i=1}^{n}\left(T_i - \overline{T}\right)^{0.5}} \right)^2 \tag{1}$$

where $T_i$ refers to the $i^{th}$ interpolated temperature value, $Tm_i$ refers to the $i^{th}$ measured temperature value, $n$ is the
number of measurements, while $\overline{Tm}$ and $\overline{T}$ represent the mean for the entire dataset of the observed and interpolated values, respectively. The coefficient of determination ($R^2$) varies from 0.0–1.0; however, the information it provides is limited because it standardizes the differences between the observed and simulated means and variances because it only evaluates linear relationships between the variables, thus it is insensitive to the additive and proportional differences between the observed and interpolated values (Legates and McCabe, 1999).

The Coefficient of Efficiency (COE) is an improvement over $R^2$ because it is sensitive to differences in the observed and interpolated means and variables (Legates and McCabe, 1999), ranging from $-\infty$ to 1.0, and obtained by:

$$COE = 1.0 - \frac{\sum_{i=1}^{n}|Tm_i - T_i|}{\sum_{i=1}^{n}|Tm_i - \overline{Tm}|} \tag{2}$$

where a value of COE=1.0 represents a perfect model, and a value of COE=0.0 indicates that the model is not better at predicting the observed values than the observed mean, while negative values indicate that the model is less effective than the
observed mean in predicting the variation in observations.

The Mean Absolute Error (MAE) is also reported in this work because it is an unambiguous and more natural measure of average error than the Root Mean Square Error (RMSE, Willmott and Matsuura (2005)) due to the bias of RMSE when large outliers are present (Legates and McCabe, 1999) and because the RMSE does not describe average error alone and its use has been discouraged (Willmott and Matsuura, 2005). The MAE is determined by:

$$MAE = \frac{\sum_{i=1}^{n}|Tm_i - T_i|}{n} \tag{3}$$

The modified Index of Agreement (IOA, Legates and McCabe (1999)) has the advantage that errors and differences are not inflated by their squared values and is computed by:

$$IOA = 1.0 - \frac{\sum_{i=1}^{n}|Tm_i - T_i|}{\sum_{i=1}^{n}\left(|T_i - \overline{Tm}| + |Tm_i - \overline{Tm}|\right)} \tag{4}$$

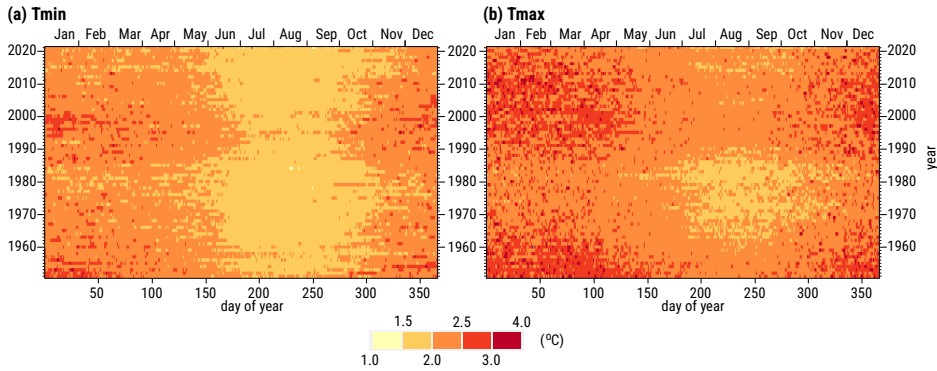

**Figure 5.** Mean Absolute Error of daily interpolations for: (a) $T_{min}$, (b) $T_{max}$.

Another advantage of the IOA is that it is related to the Mean Average Error (MAE) and the Mean Absolute Deviation
(MAD) as follows:

$$IOA = 1.0 - \frac{MAE}{MAD} \qquad (5)$$

The above metrics were computed daily for each of the three interpolated climate values through the use of the `OpenAir`
library for R (Carslaw and Ropkins, 2012). Due to the large variability of precipitation in Mexico, the Mean Absolute Error
(MAE) was only determined for $T_{min}$ and $T_{max}$ as shown in Fig. 5, where it can be seen that the MAE values are lower for $T_{min}$
than for $T_{max}$, and that for both temperatures the MAE is lower for the summer months. For $T_{min}$, the MAE varies between
1.5–2.0 °C from June through October and between 2.0–2.5 °C for the remainder months, except for some days in January
and December, where it reaches 2.5–3.0 °C. A similar behavior is observed for the MAE values of $T_{max}$, but with MAE values
0.5°C higher. The fact that the MAE exhibits lower values in summer for both $T_{min}$ and $T_{max}$ reflect a more "stable" spatial
variation of temperature for that season; this occurs because during autumn and winter Mexico experiences cold fronts, which
can not be related to elevation and affect only parts of the country.

The three dimensionless performance indices ($R^2$, COE, and IOA) obtained for the three climate variables are shown in
Fig. 6, where it can be seen that $T_{min}$ exhibits the highest values for the three performance indices; in fact, for $T_{min}$, $R^2 > 0.7$ on
most days (with $0.6 < R^2 < 0.7$ only for some days). For the same climate variable ($T_{min}$), the Index of Agreement (IOA) is $> 0.8$
for several days – in a similar (but more persistent) temporal distribution as MAE (Fig. 5) – with the remaining days having
values of $0.7 < IOA < 0.8$. Of interest is the fact that the values of the three performance indices are higher during summer, in
particular for the 1960–1990 period (as can be easily observed in Fig. 5(b)), with precipitation exhibiting a similar – but more
subtle – behaviour. The values of $R^2$, COE, and IOA for precipitation are lower due to its heterogeneity (which is why the
MAE is not shown for this variable in this first validation step). Heterogeneity of the climate variables affect the computed
performance indices, and a more dense coverage improve their estimation, as indicated by better performance values between
225 1975–1985.

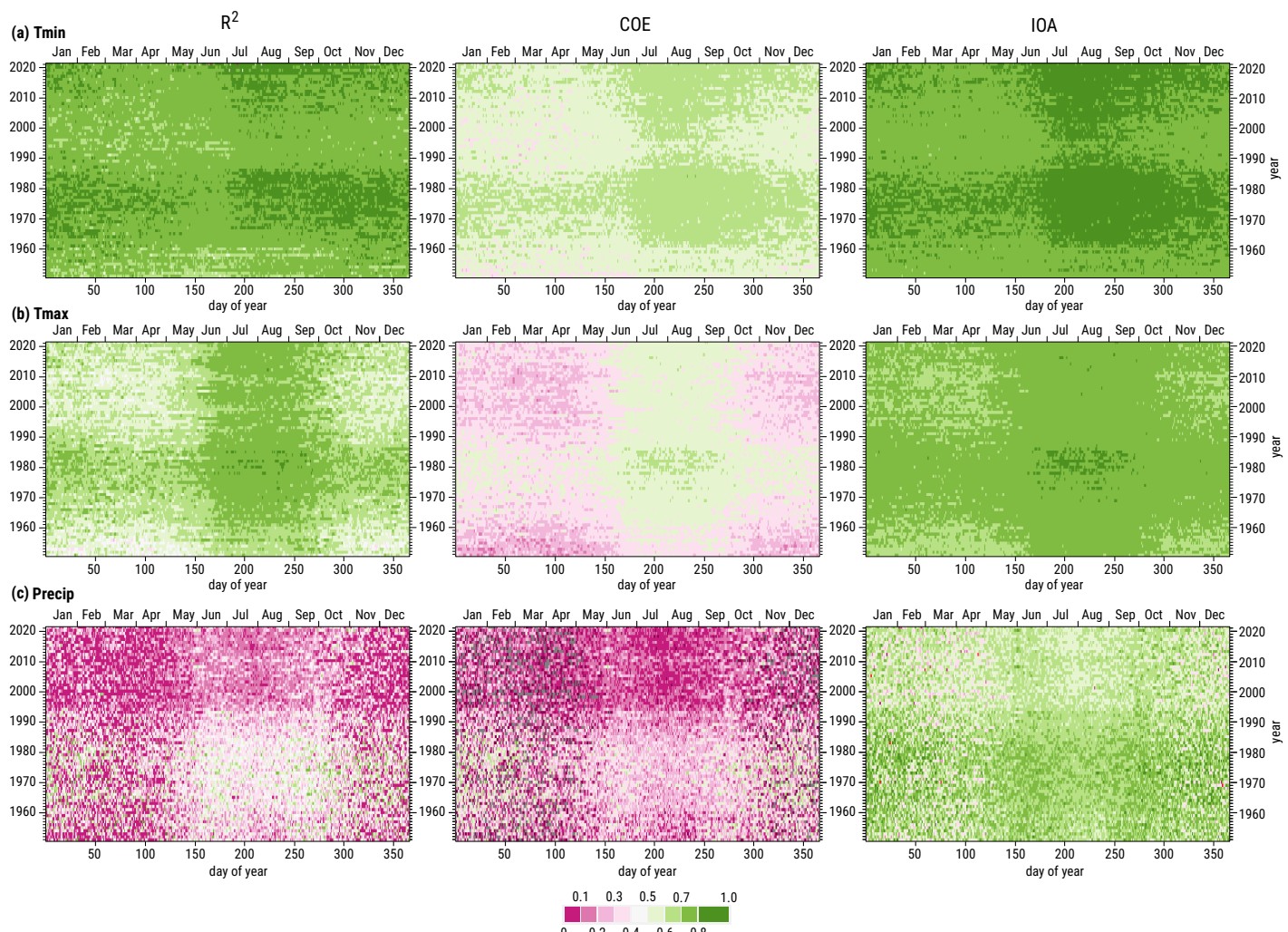

**Figure 6.** Performance indices – coefficient of determination ($R^2$), Coefficient of Efficiency (COE) and Index of Agreement (IOA) – for the daily interpolations developed in this work: (a) Minimum Temperature, (b) Maximum Temperature, and (c) Precipitation.

## 5 Comparison with other datasets and applications

Different gridded climate datasets that cover Mexico are currently available, as summarized in Table 4. These datasets have been used in Mexico for different purposes: to analyse the duration and intensity of Mexico's midsummer drought during the 1981–2010 period (Perdigón-Morales et al., 2018), to study climate trends in both the North American Monsoon (NAM) and Mid-Summer drought (MSD) regions (Cavazos et al., 2020), to analyse trends of daily rainfall indices (Colorado-Ruiz and Cavazos, 2021), and to validate the accuracy of their precipitation values on Mexico's northwestern region (Esquivel-Arriaga et al., 2024; de la Fraga et al., 2024).

Some of these datasets were developed through interpolation of observed climate values at meteorological stations (Daymet, L15) while others were derived from satellite observations (PERSIANN-CDR) or a combination of these two methodologies (CHIRPS). Due to Mexico's abrupt topography – and its effect on both temperature and precipitation – it is imperative for any method used to estimate the aforementioned climate variables in the country to consider the impact of topography on their spatial distribution. In order to validate the daily precipitation values of the HiResMexClimDB, different datasets obtained through either interpolation, remote sensing or a combination of these two are compared for five events of extreme precipitation. As can be seen in Table 4, the MexHiResClimDB has the finest spatial resolution and the longest temporal coverage of the five datasets that were compared, which are briefly described in the following paragraphs.

### 5.1 Daymet

The latest release of the Daymet database (version 4, Thornton et al. (2021)) provides daily $T_{min}$ and $T_{max}$ along with Precip and other variables for North America – Mexico, the Conterminous U. S. (CONUS) and Canada – at a resolution of 1 km for the 1980–2023 period. The estimation of $T_{min}$, $T_{max}$ and Precip in Daymet is achieved through a truncated Gaussian filter that uses inputs from several weather stations and weights that reflect the spatio-temporal relationships between each cell and the surrounding stations. For each grid cell the station list and associated weights are calculated on a yearly basis for $T_{min}$, $T_{max}$ and Precip; additionaly, this dataset also includes other secondary variables such as daylight average shortwave radiation, daily average water vapor pressure, daylength and an estimate of accumulated snowpack. The daylength estimate is based on geographic location and time of year, while the remaining secondary variables are derived from $T_{min}$ and $T_{max}$ and Precip based on atmospheric theory and empirical relationships, as detailed in Thornton et al. (2021).

### 5.2 L15

The L15 dataset (Livneh et al., 2015) is a daily gridded dataset with a resolution of $1/16°$ (3'20" or $\approx$6 km) derived from observed Precip, $T_{min}$ and $T_{max}$ for North America (Mexico, the CONUS, and regions of Canada south of 53 N$°$) for the 1950–2013 period. This dataset was created by applying the SYMAP interpolation algorithm, which uses Inverse Distance Weighting (IDW, Shepard (1984)). To develop this dataset, weather stations from Mexico, CONUS and Canada were used; however, the selection criteria on which stations to use were different according to their location: a minimum of >20 years of data were

required for weather stations located in CONUS or Canada, while only >50 days of data were required for the weather stations located in Mexico.

In order to consider the effect of topography on both temperature and precipitation, a lapse rate of 6.5°C/km was applied to $T_{min}$ and $T_{max}$, while precipitation was scaled based on existing estimates of monthly precipitation that were developed by considering topography into account. For the CONUS, the Parameter-elevation Regressions on Independent Slopes Model (PRISM, Daly et al. (1997)) dataset was used to scale precipitation, while for both Canada and Mexico the gridded climate dataset developed by Wehner et al. (2011) – which was obtained by using trivariate thin plate smoothing splines that employed latitude, longitude and elevation as predictors – was used to incorporate the effect of topography on precipitation.

## 5.3 PERSIANN-CDR

The global dataset Precipitation Estimation from Remotely Sensed Information using Artificial Neural Networks-Climate Data Record (PERSIANN-CDR, Ashouri et al. (2015)) provides world wide daily precipitation at a resolution of 0.25° ($\approx$25 km at the equator) since 1983. The PERSIANN dataset is estimated through an Artificial Neural Network (ANN) model that extracts cold-cloud pixels and neighbouring features from GEO infrared images that associates variations in each pixel's brightness temperature to estimate the pixel's rainfall rate and uses monthly data from the Global Precipitation Climatology Project (GPCP) to reduce biases in the estimation of precipitation (Ashouri et al., 2015).

## 5.4 CHIRPS

The Climate Hazards group Infrared Precipitation with Stations (CHIRPS) was developed using precipitation estimates based on infrared cold cloud duration observations calibrated through the Tropical Rainfall Measuring Mission Multi-satellite PRecipitation Analyis version 7 and a moving window regression that used latitude, longitude, elevation and slope – as detailed in Funk et al. (2015). The CHIRPS dataset provides world wide daily precipitation since 1981 with a spatial resolution of 3' ($\approx$5.4 km at the equator). Although the main advantage of precipitation estimates derived from satellite observations is their aereal coverage, it should be kept in mind that the main drawback of estimates based on IR techniques (PERSIANN-CDR and CHIRPS) is that they are based on the assumption that colder clouds produce more intense rainfall and may miss heavy precipitation from shallow clouds (Behrangi et al., 2016).

## 5.5 Validation and comparison of Precipitation

The spatial variation of daily precipitation from the MexHiResClimDB dataset is compared with the previously mentioned datasets using five different events of extreme precipitation in Mexico (shown in Fig. 7 and caused by hurricanes or tropical storms). These events and their impact in Mexico are briefly described in the following paragraphs:

1. On September 16th of 1988, Hurricane Gilbert – once labelled the "storm of the century" because of the meteorological records it set (Meyer-Arendt, 1991) – caused torrential rains, which led to floods that caused fatalities and destruction in Monterrey, Mexico's third largest city (Meyer-Arendt, 1991). The first landfall of this hurricane was in Cozumel on

**Table 1.** Spatial resolution and temporal coverage of the MexHiResClimDB and the other four gridded climate datasets used for comparison.

| Dataset | Source | Climate variable | Spatial resolution | Coverage Temporal | Area |
|---|---|---|---|---|---|
| MexHiResClimDB (Carrera-Hernández, 2025a) | Weather station | $T_{min}$, $T_{max}$, Precip | 20" ≈ 0.6 km | 1951 – 2020 | Mexico |
| Daymet (Thornton et al., 2021) | Weather station | $T_{min}$, $T_{max}$, Precip + 4 more* | 35" ≈ 1.0 km | 1980 – 2023 | North America |
| L15 (Livneh et al., 2015) | Weather station | $T_{min}$, $T_{max}$, Precip | 3'20" ≈ 6.0 km | 1950 – 2013 | North America |
| CHIRPS (Funk et al., 2015) | Satellite and weather station | Precip | 3'00" ≈ 5.4 km | 1981 – 2023 | Global |
| PERSIANN CDR (Ashouri et al., 2015) | Satellite | Precip | 15'00" ≈ 25.0 km | 1983 – 2024 | Global |

*The other three variables included in Daymet are shortwave radiation, vapor pressure, snow water equivalent and day length.

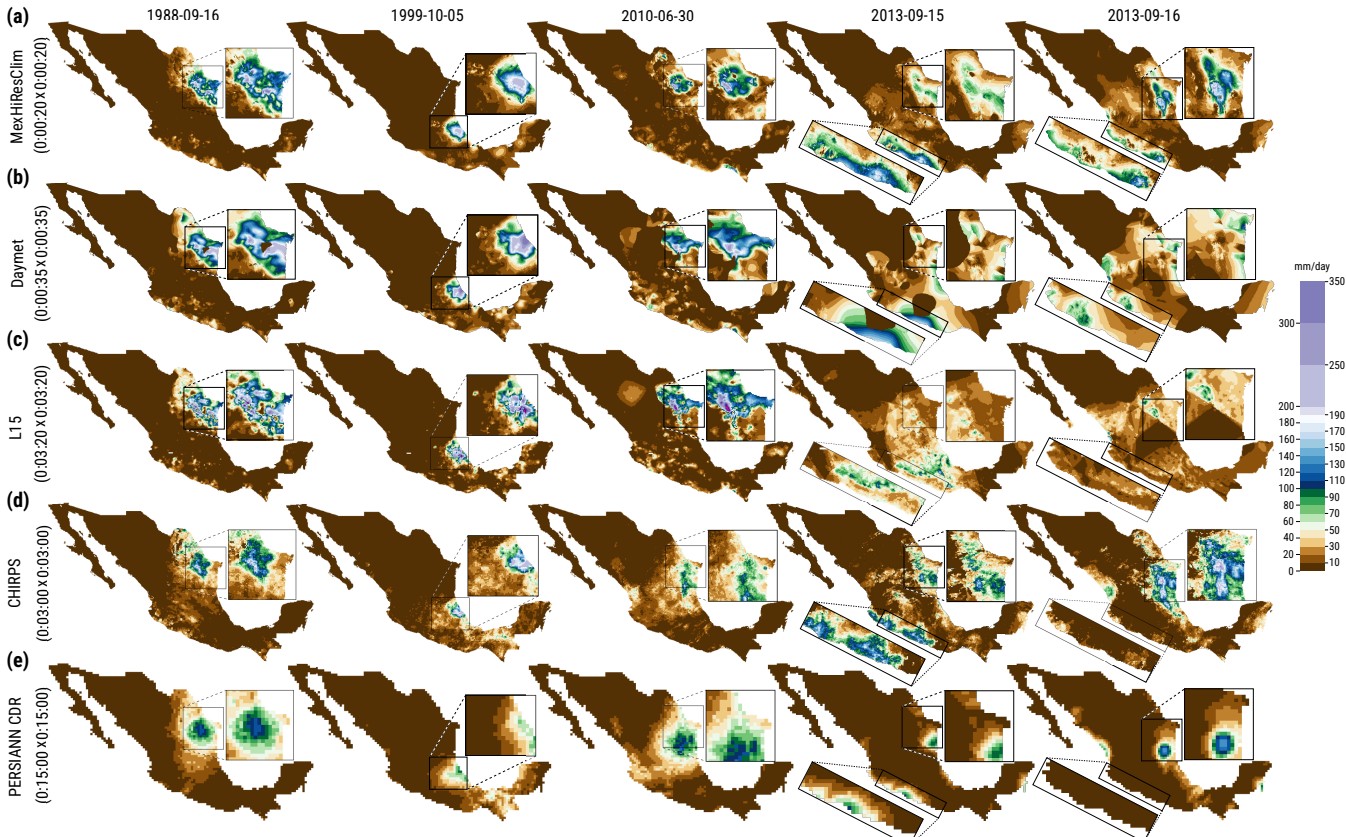

**Figure 7.** Spatial distribution of daily precipitation on five different days according to five different datasets: (a) MexHiResClim, (b) Daymet, (c) L15, (d) CHIRPS, and (e) PERSIANN CDR. These days were selected due to the large precipitation registered on them: September 16th, 1988, when Hurricane Gilbert caused large precipitation events – and flooding – in the Monterrey Metropolitan Area (MMA); October 5th 1999, when Tropical Depression 11 caused several landslides in the state of Puebla; June 30th 2010, when Hurricane Alex caused flooding in the MMA; September 15–16 of 2013, when Tropical storm Manuel in the Pacific and Hurricane Ingrid (Cat 1) in the Gulf of Mexico occurred simultaneously, causing floods – and triggering landslides – in different parts of Mexico. Only the MexHiResClim dataset adequately represents the precipitation caused in both the Pacific and the Gulf of Mexico regions for September 15–16 of 2013.

September 14th as a Force 5 Hurricane, and due to the floods it caused, 225 people died in the Monterrey Metropolitan Area (Aguilar-Barajas et al., 2019).

2. The large rainfall events of October 4–6 of 1999 caused by Tropical Depression 11, triggered several landslides in the northern Sierra of Puebla, affecting different areas (Capra et al., 2003a, b; Alcántara-Ayala, 2004; Borja-Baeza and Alcántara-Ayala, 2004; Alcántara-Ayala et al., 2006). A total of nearly 3000 landslides that ranged from soil slides to debris flows and avalanches occured in this area and on October 5, in the town of *Teziutlán*, a single landslide caused approximately 150 casualties in this town – with a total of 263 casualties – affecting 1.5 million people.

3. In July 2010 Hurrican Alex affected the Monterrey Metropolitan Area. Hurricane Alex was a Category 2 hurricane before landfall in Tamaulipas (Cázares-Rodríguez et al., 2017) and flash floods triggered by it caused fifteen fatalities in the Monterrey Metropolitan Area (Aguilar-Barajas et al., 2019).

   4. The simultaneous ocurrance of the tropical storm Manuel on the Pacific coast and of hurricane Ingrid (Cat 1) on the Gulf of Mexico in September 2013 caused flooding and substantial damage in several states of Mexico. The rainfall
from tropical storm Manuel triggered a landslide at the *La Pintada* village in Guerrero, causing 78 fatalities – with an additional eight missing people (Alcántara-Ayala et al., 2017). The floods caused by Manuel destroyed several highways and bridges in the state of Guerrero, leaving 40,000 tourists stranded in the Acapulco bay because its main highway could not be used for a week, and its airport was flooded (CENAPRED, 2014).

   The spatial distribution of precipitation for each of the aforementioned events is shown in Fig. 7, where it can be seen that
PERSIANN-CDR exhibits lower precipitation values than the other four datasets for the five events considered, and that the areal coverage of precipitation obtained with both CHIRPS and PERSIANN CDR is smaller than the one obtained with the other three datasets (which can be clearly seen for the events of 1988, 1999 and 2010). Of particular interest are the events of 2013: for September 15th, the MexHiResClimDB shows a large area with precipitation along the Pacific Coast caused by the Tropical Storm Manuel and another area with precipitation on the Northeastern region of Mexico, near the Gulf of Mexico's
coast, caused by Hurricane Ingrid. This precipitation pattern is similar to that reported in Pedrozo-Acuña et al. (2014) and Rosengaus-Moshinsky et al. (2016), but not present on the L15 and Daymet datasets (as shown in Fig. 7(b) and (c)). In fact, for this day (2013-09-15), the L15 dataset shows larger precipitation on the leeward side of the mountain range near Acapulco (Fig. 1(a)) – which is opposite to what actually occured, and represented on the other four datasets, although with some caveats, as is the case of Daymet (i.e. interpolation artifacts).

To provide further insight into how well these five datasets represent the registered precipitation of these five events, the Coefficient of Determination ($R^2$), the Coefficient of Efficiency (COE) and the Index of Agreement (IOA) were determined for each event and dataset by leave-one-out cross-validation in the case of the MexHiResClimDB and through raster sampling of the remainder datasets using the `v.what.rast` command of the GRASS GIS (which updates the table attributes of a vector map, which in this case it consisted of the weather stations used to undertake $KED_l$ for each day). These performance
statistics are shown in Fig. 8 along with their respective scattergrams of differences; however, it should be kept in mind that

the performance metrics shown in the aforementioned figure for the MexHiResClimDB can not be directly compared with the metrics of Daymet, L15 or CHIRPS, because for the latter three cases some of the weather stations used to compute the metrics were used to develop the datasets – thus, the performance metrics obtained through cross-validation for the MexHiResClimDB are expected to be lower. However, as can be seen on Fig. 8, the PERSIANN CDR is the dataset with the lowest metrics for four of the five events considered, followed by CHIRPS. For the precipitation events considered, L15 showed the largest performance values for only one event (1988-09-16), while Daymet for two (1999-10-05, 2010-06-05) and MexHiResClimDB for the remaining two events (2013-09-15, 2013-09-16). The differences between the performance statistics of L15 or Daymet compared to MexHiResClimDB for the first three events are not drastic: if the IOA is considered, for 1988-09-16, $IOA_{L15} = 0.863$, while $IOA_{Daymet} = 0.811$ and $IOA_{MexHiResClimDB} = 0.751$. For the last two days – for which the performance indices of the MexHiResClimDB are better – these differences are larger, as for 2013-09-15, $IOA_{MexHiResClimDB} = 0.703$, $IOA_{Daymet} = 0.525$ and $IOA_{L15} = 0.334$. However, if the Coefficient of Efficiency (COE) is considered, the performance difference for the 2013 events is even larger, as $COE_{L15} = -0.331$, $COE_{Daymet} = 0.051$ and $COE_{MexHiResClimDB} = 0.405$. From this analysis, it can be concluded that the precipitation patterns obtained with the MexHiResClimDB represent precipitation in a better way than the other four datasets considered in this work.

### 5.5.1 Extreme records of precipitation

The MexHiResClimDB provides not only daily precipitation, but monthly and yearly aggregated values as well. One application of this new database is the generation of a summary of precipitation extremes (i.e. wettest and driest) at the aforementioned aggregation times (which will provide information that is not available for Mexico at this time). In order to generate this information, the volume of precipitation was computed for each day and aggregated at the national level (the use of volume was selected instead of mm due to the variability of precipitation in Mexico, as shown in Fig. 7). In order to create this information, the `r.univar` command of the GRASS GIS – which computes univariate statistics (such as minimum, maximum or total sum) from the non-null cells of a raster map – was used to compute the nation-wide precipitation volume at the required aggretion time and stored in a relational database (PostgreSQL) for efficient handling (which is required for monthly and daily data). With this procedure, the ten wettest and driest days, months and years were obtained and summarized in Table 2, where it can be seen that the wettest day was 1970-09-26 ($28.878 \times 10^9$ m$^3$) – which surprisingly was not caused by a hurricane – while September of 1974 had the largest precipitation events for two and three consecutive days (21–22 and 20–22 respectively), caused by the Fifi-Orlene Hurricane (which entered as Tropical Storm on the eastern side of Mexico and moved westwards over the country to regain energy once entering the Pacific Ocean to turn into Hurricane Orlene). Of interest is the 10th wettest day (2010-02-03), which occurred in February (outside Mexico's rainy season). The month with the largest precipitation was September of 2013 ($387.716 \times 10^9$ m$^3$, caused by both Manuel and Ingrid), while the wettest year was 1958 ($1,826.167 \times 10^9$ m$^3$). The driest year was 1953 while January of the same year was the driest month. An interesting finding was that the precipitation event of 2013-09-16 ($25.952 \times 10^9$ m$^3$) is ranked as the 6th wettest day and that no wettnes or dryness tendency is easily seen in Table 2.

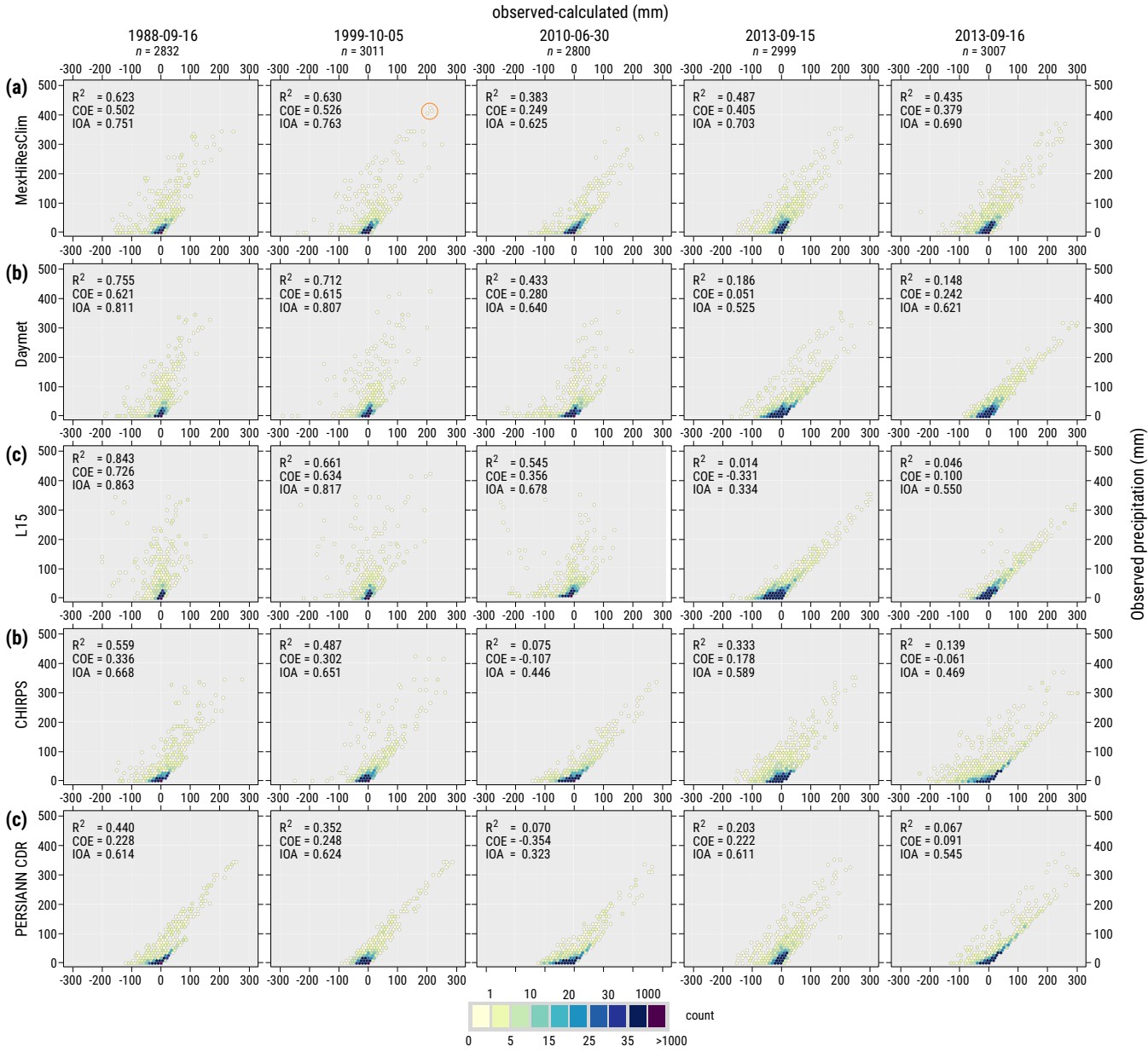

**Figure 8.** Validation scatterplots and error metrics for the precipitation events shown in Figure 7 for (a) MexHiRexClim, (b) Daymet, (c) L15, (d) CHIRPS, and (e) PERSIANN CDR. It is important to keep in mind that the values shown for Daymet and L15 do not correspond to cross-validation, but rather to sampling of the rasters provided by each dataset; accordingly, cross-validation of both Daymet and L15 would exhibit lower values for the performance indices used. It can be seen that MexHiResClim is the only dataset that adequately represents the precipitation events of September 15–16 caused by the presence of Tropical Storm Manuel and Hurricane Ingrid.

**Table 2.** Countrywide (a) Maximum and (b) Minimum values of daily, monthly and yearly accumulated values for precipitation.

| Maximum | | | | | | Minimum | | | | | |
|---|---|---|---|---|---|---|---|---|---|---|---|
| daily | | monthly | | yearly | | daily | | monthly | | yearly | |
| date | precip $(10^9 \text{ m}^3)$ | date | precip $(10^9 \text{ m}^3)$ | date | precip $(10^9 \text{ m}^3)$ | date | precip $(10^9 \text{ m}^3)$ | date | precip $(10^9 \text{ m}^3)$ | date | precip $(10^9 \text{ m}^3)$ |
| 1970-09-26 | 28.878 | 2013-09 | 387.716 | 1958 | 1,826.167 | 1951-01-17 | 0.000 | 1953-01 | 6.129 | 1953 | 1,106.309 |
| 1967-09-22 | 27.725 | 2010-07 | 379.970 | 2013 | 1,646.250 | 1953-02-05 | 0.000 | 1984-04 | 8.882 | 1957 | 1,158.624 |
| 1973-06-22 | 26.975 | 1958-09 | 355.837 | 1981 | 1,591.904 | 2007-03-01 | 0.001 | 1955-04 | 8.910 | 1994 | 1,161.331 |
| 1974-09-21 | 26.763 | 1955-07 | 354.232 | 1984 | 1,552.483 | 1956-03-05 | 0.001 | 1998-04 | 9.381 | 1951 | 1,164.111 |
| 1974-09-22 | 26.333 | 1955-09 | 351.492 | 1978 | 1,532.719 | 1977-03-13 | 0.001 | 1975-04 | 9.427 | 2011 | 1,164.234 |
| 2013-09-16 | 25.952 | 1973-08 | 349.515 | 1992 | 1,524.137 | 1953-01-05 | 0.001 | 1960-03 | 9.699 | 1962 | 1,176.275 |
| 2013-09-15 | 24.925 | 1978-09 | 339.886 | 2015 | 1,522.222 | 1953-01-10 | 0.002 | 1955-03 | 10.246 | 1987 | 1,179.864 |
| 1974-09-20 | 23.987 | 2014-09 | 338.307 | 1976 | 1,517.643 | 1956-03-04 | 0.002 | 1962-02 | 10.497 | 1982 | 1,191.108 |
| 1978-09-22 | 23.954 | 1976-07 | 334.457 | 2010 | 1,517.590 | 1952-02-13 | 0.003 | 1984-03 | 10.934 | 2009 | 1,192.888 |
| 2010-02-03 | 23.818 | 1969-08 | 333.554 | 1955 | 1,509.553 | 1958-02-27 | 0.003 | 1970-04 | 11.447 | 1956 | 1,201.528 |

### 5.5.2 Validation with independent data from Automatic Weather Stations

The comparison undertaken in the previous section used cross-validation for the MexHiResClimDB and sampling at the location of conventional weather stations for the other gridded datasets. To improve the validation – and intercomparison – of the gridded climate datasets, independent data from different Automatic Weather Stations (AWSs) operated by Mexico's National Weather Service (SMN) are used in this section. The AWSs acquire data every 10 minutes and these data are distributed by the SMN after being requested, with the data being provided in one spreadsheet file per month, which had
to be proccessed in a similar way to that shown in Fig. 2. For the validation reported in this section, the two wettest months (September 2013 and June 2010, Table 2) were selected in order to have the maximum number of days with precipitation. Once the data from the AWSs were imported into PostgreSQL, stations with full data (registered every ten minutes) had to found; the original idea was to use the same stations for both months, but this proved to be unfeasible due to the lack of continuous data. The stations that were selected for this validation are shown in Fig. 9, where it can be seen that they are located at different
elevations (with station 3102 found at an elevation of 4 meters in the Yucatán Península, while station 1504 is found at 4077 meters) and throughout Mexico (except Baja California, due to the lack of data on the AWSs located there). It can be noted that stations 501, 502 and 810 are very close to the border between Mexico and the U.S., and are useful to explore the impact of interpolation near Mexico's northern limit, because the MexHiResClimDB was developed by only using data from Mexico, whereas Daymet and L15 used weather stations from Mexico, the U.S. and Canada, or globally (CHIRPS).
The ten-minute precipitation data recorded at each of the AWS was aggregated at a daily level through the use of the `subdaily2daily` function of the `hydroTSM` library for R (Zambrano-Bigiarini, 2024) for September 2013 and July 2010 (61 days) and their corresponding mass curve was also computed in order to compare it with the precipitation mass curve obtained from the different gridded precipitation datasets, as shown in Figures 10(a) and (b); for this analyses the PERSIANN dataset was not considered due to its low spatial resolution. The acquisition time of data from Mexico's conventional weather

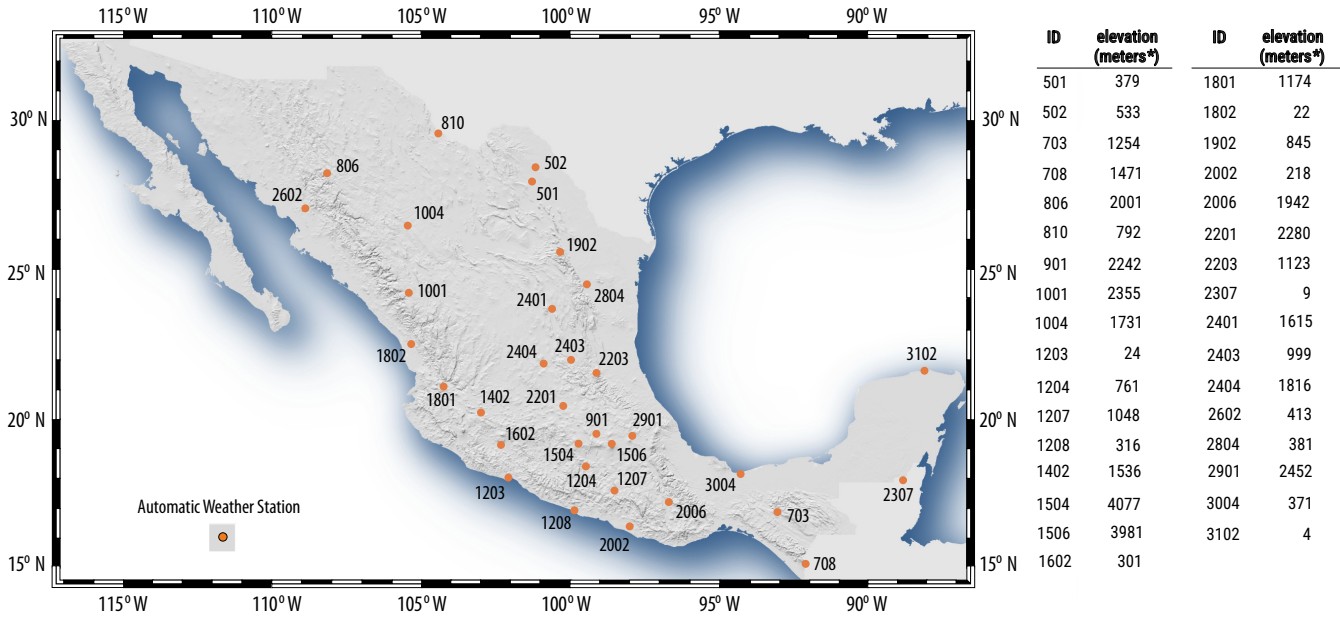

**Figure 9.** Location of Automatic Weather Stations used for further validation.

stations is 8:00 a.m., while the timestamp (i.e. date and time) from AWS data is in UTC time; this required to modify the acquisition time of the AWS data to match the acquisition time of the conventional stations (which was done in PostgreSQL).

It was decided to use the precipitation mass curve in order to show the difference in accumulated precipitation throughout and at the end of the month for each case. Of interest is the fact that the majority of the mass curves shown in Figures 10(a) and (b) show a similar behaviour to that of the AWS. In addition, the location of the AWS used also showed the limitations of the spatial resolution used in both L15 and Daymet for September 2013, as they do not provide data for the AWS 2307 (Fig. 10(a) – located near the border to Belize, Fig. 9) or for July 2010 in the case of L15, which could not provide data for the AWS 3102 (Fig. 10(b) – located at the north of the Yucatán Peninsula, Fig. 9).

For the case of September 2013, the MexHiResClimDB provided a lower MAE for 17 of the 20 cases shown in Fig. 10(a) and for 7 of the 20 cases shown for July 2010 (Fig. 10(b)). Of interest is the precipitation registered at AWS 1208 in September 2013, as the monthly precipitation registered by this station was around 1100 mm, while the monthly value obtained with data from MexHiREsClimDB was around 600 mm (and 200 mm in the case of L15). For this AWS, the mass curves for all gridded datasets show a clear underestimation of precipitation for the events of September 14 and 15. This underestimation could have been caused by the fact that these events were caused by Tropical Storm Manuel and that the data registered at the conventional station (used on all the gridded datasets being compared) were wrong (i.e. lack of accesibility to the weather station or damage caused by strong winds). This situation contrasts with that of AWS 2203, which registered a total of 1025 mm, but which was represented similarly – although overestimated (1150 mm) – by MexHiResClimDB, while the monthly total by CHIRPS at this station was around 550 mm, and of only 50 mm by L15 Fig. 10(a).

Of particular interest are the mass curves of AWS 501 for September 2013, because this station is very close to the Mexico-U.S.A. border. At this station, the precipitation mass curve with the lowest MAE was the one obtained with precipitation from the MexHiResClimDB (MAE = 28.2 mm) followed by CHIRPS (MAE = 29.6 mm), L15 (MAE = 60.1 mm) and Daymet (MAE = 76.2 mm). Unfortunately, there was no data for station 502 on this date, but both stations have data for July 2010, when Hurricane Alex caused large precipitation events – and flooding – in northeastern Mexico. As can be seen on Fig. 10(b), the large precipitation events registered at station 501 and 502 for July 2010 were underestimated by all gridded datasets, in particular for station 501, which registered nearly 500 mm in 9 days, while the precipitation from MexHiresClimDB was nearly 300 mm (MAE = 181 mm), followed by Daymet (160 mm, MAE = 288 mm), CHIRPS (120 mm, MAE = 329 mm) and L15 (100 mm, MAE = 349 mm). For station 502 (which is closest to the border between Mexico and the U.S.A), data obtained from the MexHiResClimDB represented in a better way the precipitation registered at the AWS (with a MAE = 40.4 mm while the MAE obtained for the remainder gridded datasets was above 200 mm).

While Figure 10 compares the monthly precipitation mass curves between the registered values at the AWSs and the different gridded datasets for September 2013 and July 2010, the daily comparison between these datasets is shown in Figure 11 along with their corresponding performance metrics. For the 61 days sampled at the location of the AWSs listed in Fig. 10, the lowest MAE (8.7 mm) was obtained with precipitation data from the MexHiResClimDB, followed by L15 (9.5 mm), Daymet (10.1 mm) and CHIRPS (11.7 mm). As can be seen on Fig. 11, all gridded datasets tend to underestimate daily precipitation above 50 mm, which can also be observed on Fig. 8 when daily data from the traditional weather stations were used for the dataset comparison. Of note is how both Daymet and L15 could not be sampled at some stations due to their spatial extent and resolution and because L15 has an areal data gap where Lake Chapala – Mexico's largest lake (Carrera-Hernandez, 2018) – is located. This is why the hex-bin scatterplots of these datasets exhibit a lower number of points (1160 and 1098 for Daymet and L15, respectively).

### 5.5.3 Monthly precipitation values at national and watershed levels

To conclude the comparison between the gridded precipitation datasets, this section compares the nation-wide accumulated monthly precipitation for both September 2013 and July 2010, while computing the corresponding monthly value for basins where one of the AWSs used in the precipitation mass curve analysis was found. The watersheds were obtained from the recently developed Mexico's Watershed Database (MexWatDB, Carrera-Hernandez (2025)), which consists of a nested classification sytem for Mexico based on the Pfafstter organization scheme. These watersheds were selected in order to show the spatial variability of climate variables inherent to the spatial resolution of each dataset.

The accumulated monthly precipitation (at both national and basin levels) for September 2013 and July 2010 are shown in Figures 12 and 13 where it can be seen that Daymet and L15 show interpolation artifacts in large areas of northern Mexico – in particular for September 2013. At the national level, the MexHiResClimDB provides the largest volume of precipitation ($387.16 \times 10^9$ m$^3$), while with L15 the computed volume is of $309.789 \times 10^9$ m$^3$. At the watershed level, for the basin where AWS 708 is found (with an area of 330 km$^2$), the volume obtained with MexHiResClimDB is $203.9 \times 10^6$ m$^3$, which is four times the volume obtained with L15 ($49.4 \times 10^6$ m$^3$); this basin corresponds to the *Cahoacán* river, which has a steep slope,

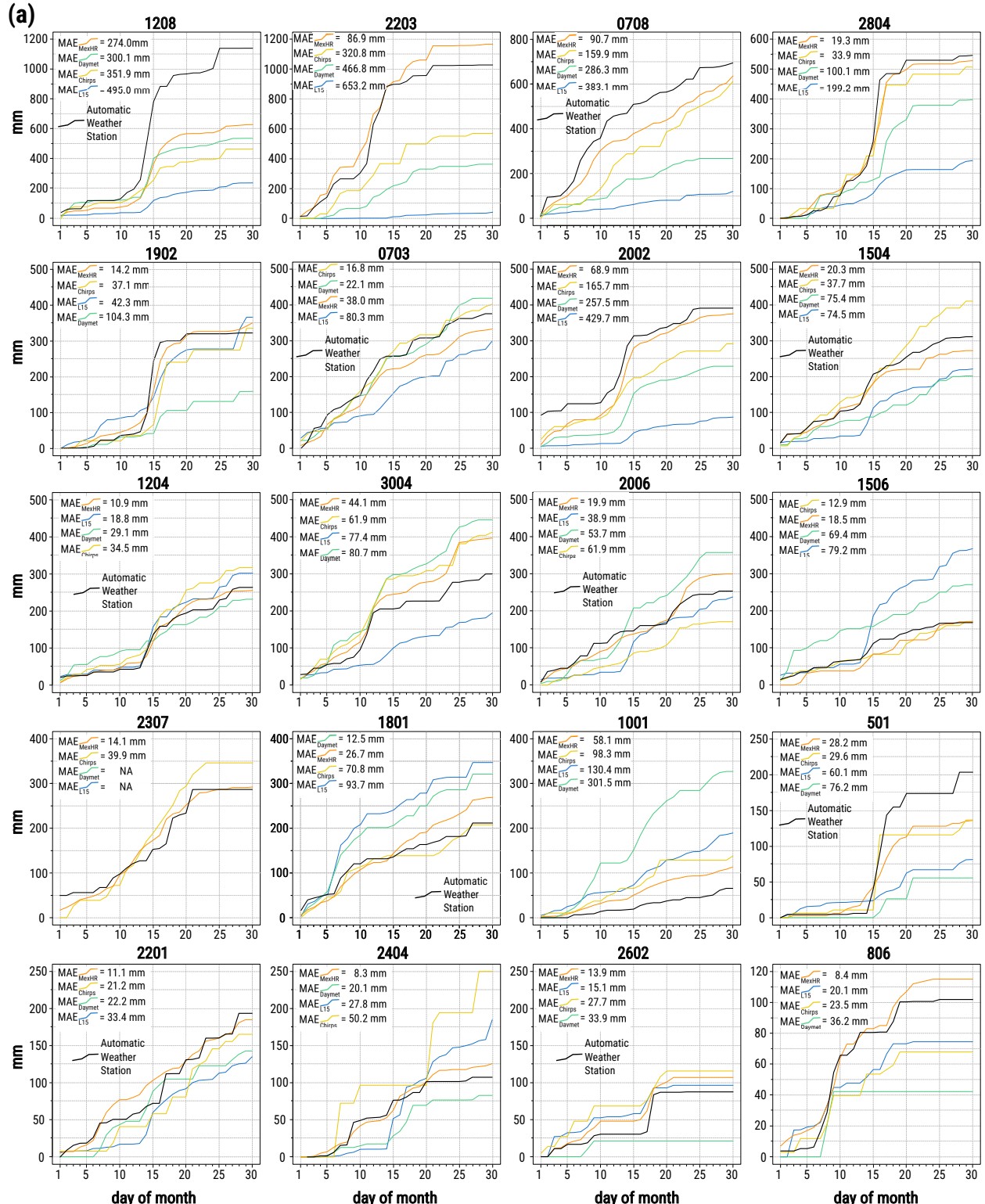

**Figure 10.** Comparison of precipitation mass curves obtained with data from Weather Stations and MexHiResClimDB, Daymet, L15 and CHIRPS for (a) September 2013, and (b) July 2010. Note the change in vertical scale for all plots, as it was modified in order to show the differences between the precipitation mass curves. For July 2010, the L15 could not be sampled because this dataset has an areal data gap where Lake Chapala – Mexico's largest lake (Carrera-Hernandez, 2018) – is located.

**(b)**

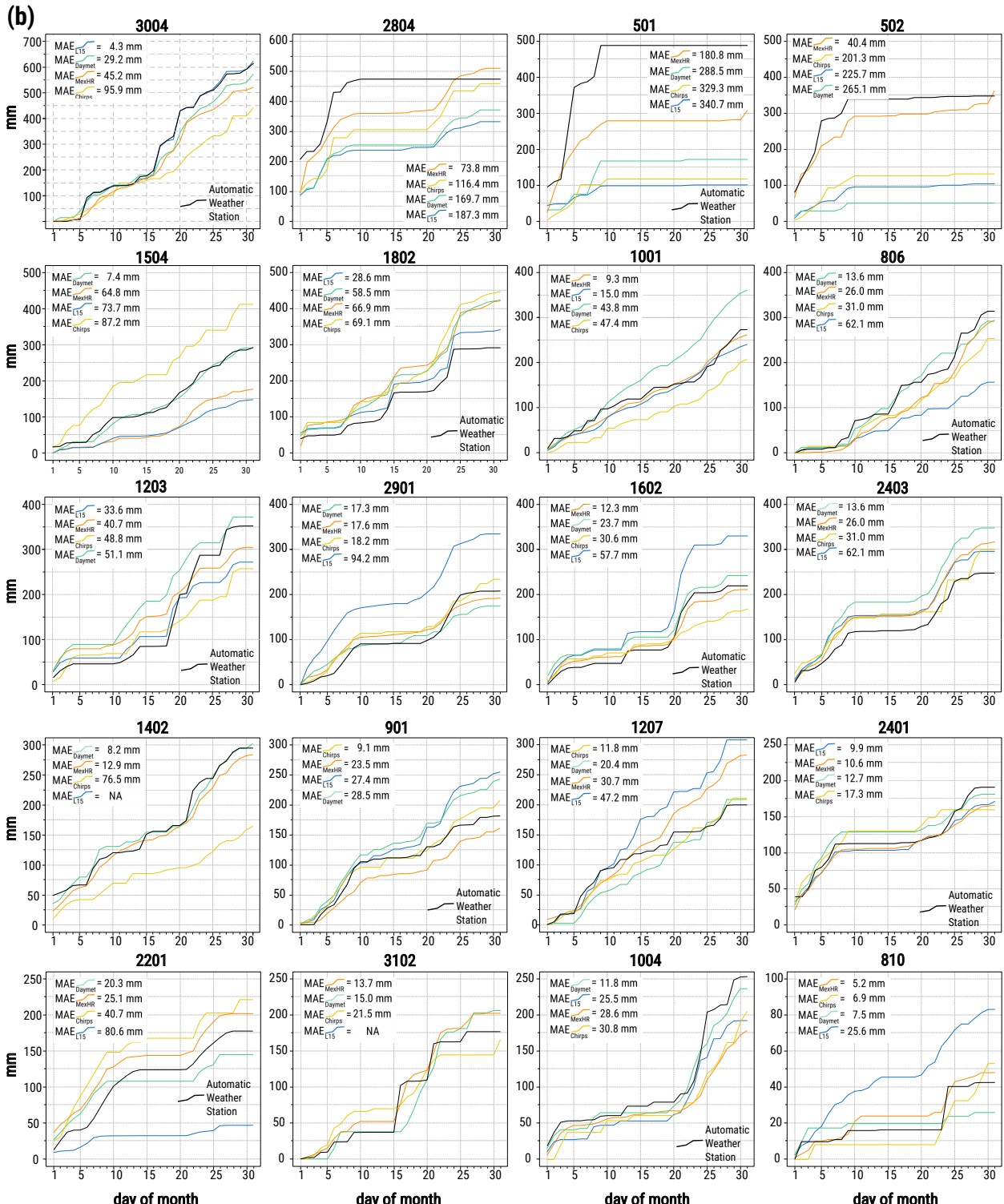

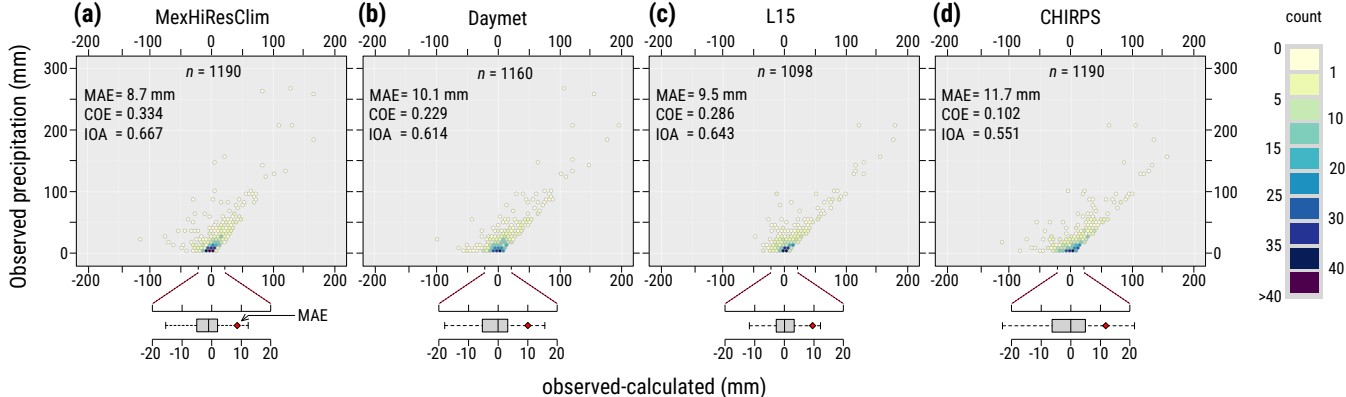

**Figure 11.** Comparison between daily values observed at the Automatic Weather Stations and gridded daily precipitation values for (a) MexHiResClimDB, (b) Daymet, (c) L15, (d) CHIRPS. The number of samples is lower for Daymet and L15 because they could not be sampled at the location of AWSs 2307 and 3102 due to their spatial resolution.

discharges on the Pacific Ocean and constantly floods the city of *Tapachula*. The AWS 0708 is found in the upper region of this basin, at an elevation of 1471 meters, and the monthly accumulated precipitation registered at the AWS (700 mm) was larger than that obtained with MexHiResClimDB (625 mm) and CHIRPS (600 mm). Of interest is the impact of the spatial variation of precipitation in this small basin, because although the difference between the monthly precipitation obtained with MexHiResClimDB and CHIRPS is of only 25 mm (Fig. 10(a)) the difference in precipitated volume is of nearly 30% (203.9 and $141.9 \times 10^6$ m$^3$ respectively).

The spatial distribution of precipitation for July 2010 (Fig. 13) near the Mexico-U.S.A. border – where AWS 502 is located – shows the large difference in precipitation obtained with the four different datasets considered. As can be seen on Fig. 10(b), the AWSs 501 and 502 registered large precipitation events for the first nine days of July 2010, due to Hurricane Alex, and flooding the city of *Piedras Negras*, near the outlet of the *Escondido* river and whose watershed corresponds to the one where AWS 502 is located. As previously mentioned, the MexHiResClimDB was the dataset that better represented the precipitation mass curve at this location, and the precipitation volumes obtained for the *Escondido* river watershed vary drastically between the gridded datasets ($737.9 \times 10^6$ m$^3$ for the MexHiResClimDB, and less than $375 \times 10^6$ m$^3$ for the remainder datasets).

## 5.6   Validation and comparison of temperature

The MexHiResClimDB also includes temperature data, and the analysis of extreme values of temperature is useful in climate change analysis; however, there is currently no information available on the coldest or hottest days in Mexico for the 1951–2020 period. To provide this information, the temperature rasters generated in the MexHiResClimDB were processed in a similar way as was done with precipitation in order to summarize the ten hottest and coldest days, months and years for $T_{min}$, $T_{max}$ and also $T_{avg}$ (which is a derived product of the interpolated temperature values), as shown in Table 3. In contrast to the precipitation values shown on Table 2 (which showed no particular trend), the highest temperature values (Table 3(a)) show a

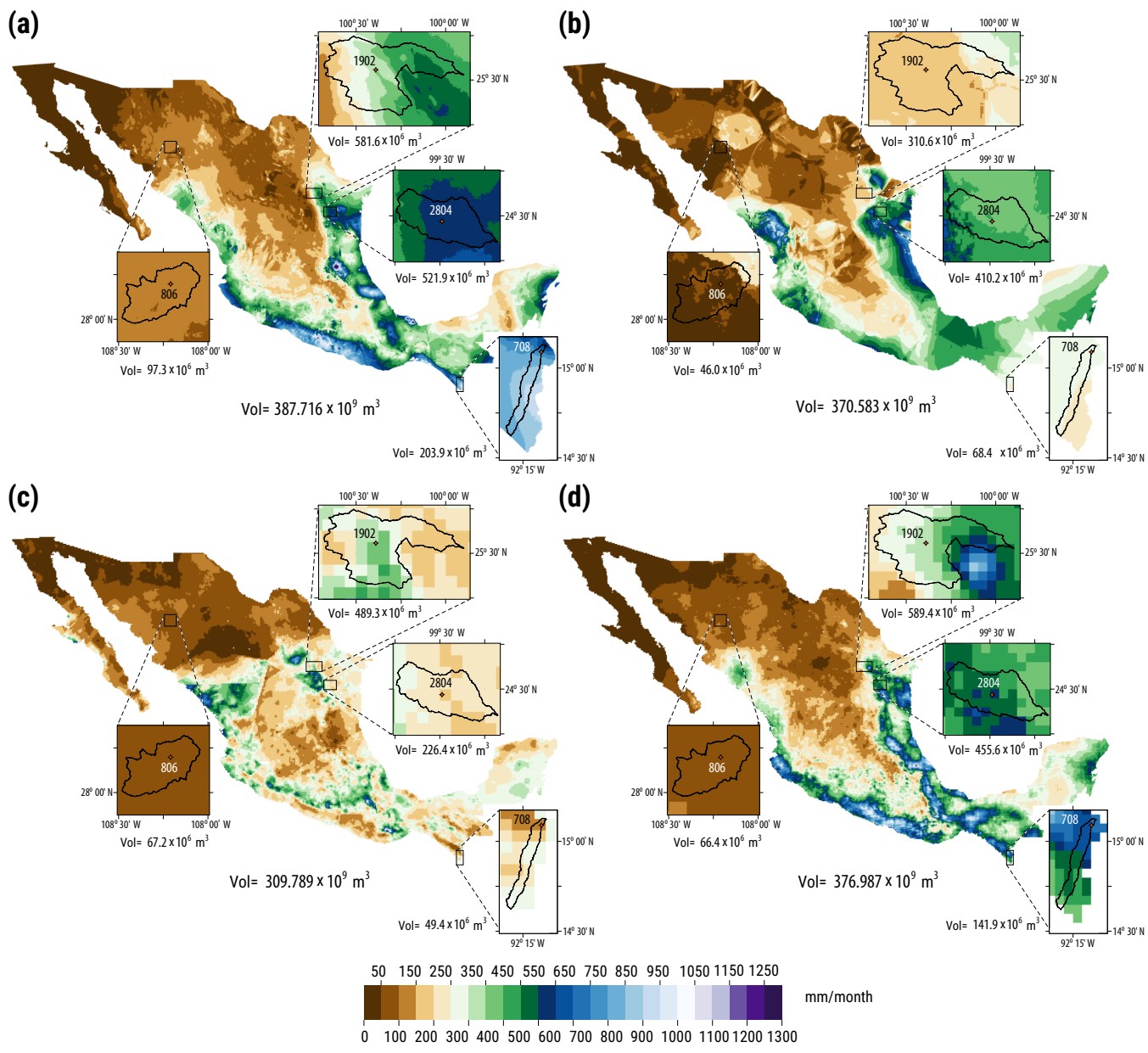

**Figure 12.** Monthly precipitation for September 2013 obtained by aggregating the daily gridded precipitation values of (a) MexHiResClimDB, (b) Daymet, (c) L15 and (d) CHIRPS. The precipitation volumes shown on the insets correspond to the monthly volume for each basin. Of note is the underestimation of precipitation for this month on Mexico's southwestern coast by both the Daymet and L15 datasets.

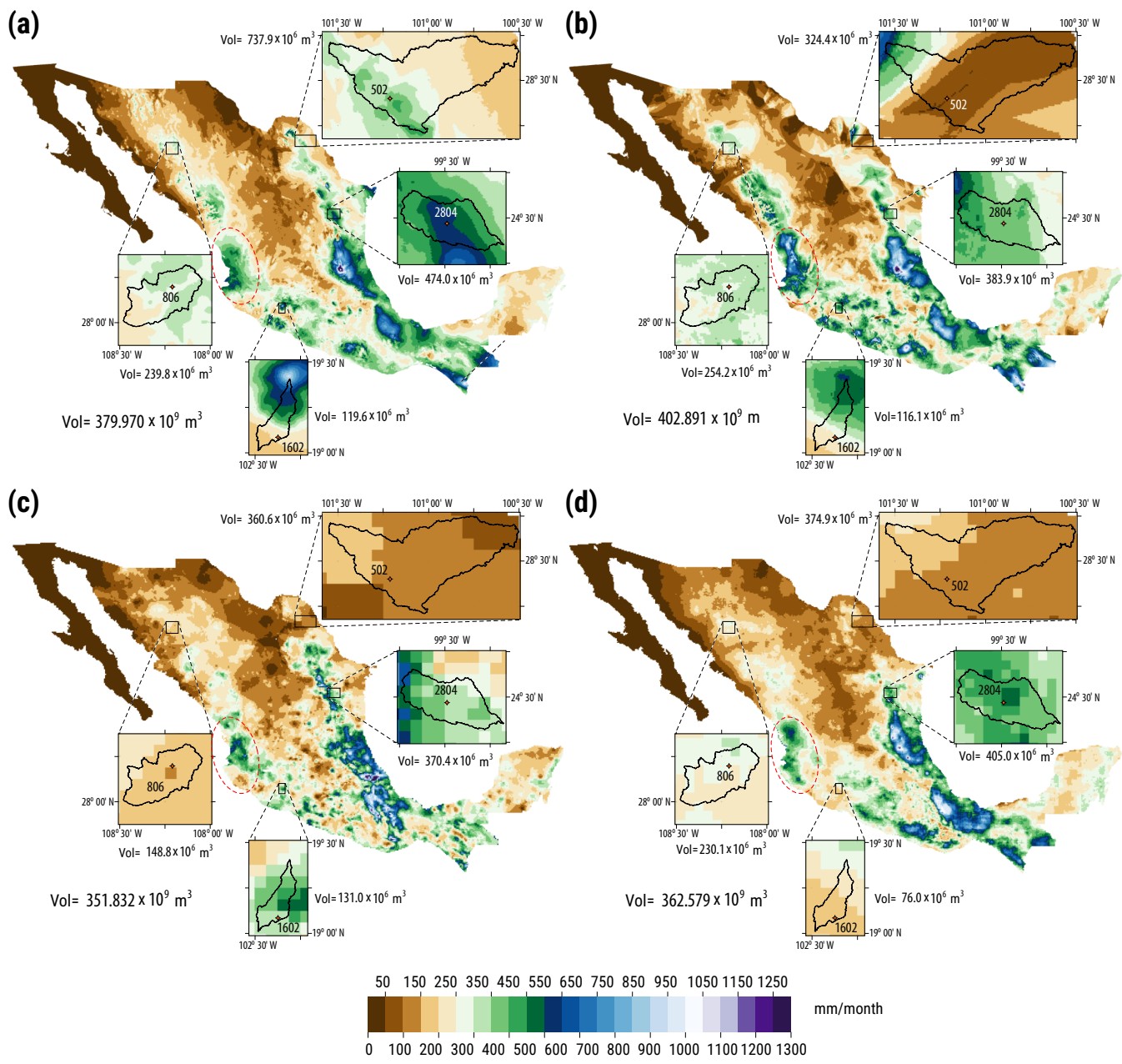

**Figure 13.** Monthly precipitation for July 2010 obtained by aggregating the daily gridded precipitation values of (a) MexHiResClimDB, (b) Daymet, (c) L15 and (d) CHIRPS. The precipitation volumes shown on the insets correspond to the monthly volume for each basin. Of note is the overestimation of precipitation by Daymet on the area highlighted by the red ellipse.

clear warming trend (in particular for minimum temperature): eight out of the ten days with the highest $T_{min}$ occurred in 2020, the two months with the highest $T_{min}$ were July and August of 2020 and the six years with the highest $T_{min}$ were 2015–2020. The hottest day was 1998-06-15, while June of 1998 was the hottest month and 2020 the hottest year; of interest is to note

that the four hottest years occurred between 2011–2020. The values of Table 3(b) show that the coldest day for the 1951–2020 period was January 12th of 1962, while the coldest month was January of 1987, with 1987 being the coldest year – as can be seen, the 1970–1979 decade was the coldest for the 1951–2020 period – and surprisingly, 2010 was the third coldest year.

To validate the interpolated temperature maps, the maximum and minimum values of $T_{max}$ (1998-6-15, 1967-1-10) and $T_{min}$ (2020-8-31, 1962-1-12) were selected to report their validation in detail. The results obtained with the leave-one-out cross-

validation are shown on Fig. 14, where it can be seen that the performance statistics are better for $T_{min}$ than for $T_{max}$ – a fact that was previously pinpointed in the previous section. The scattergrams of Fig. 14 only show the differences obtained through cross-validation for the MexHiResClimDB and not the other datasets because it is not possible to obtain cross-validation values for L15 or Daymet – and using the sampled values would yield values that are not comparable to those shown on Fig. 14. To overcome this issue, the Automatic Weather Stations (AWSs) used in the validation of precipitation data were also used for

temperature; however, not all of the AWSs shown on Fig. 10 had a full record of temperature for September 2013 or July 2010 and other stations had to be selected. As done with precipitation data, the 10-minute registered acquisition time had to be modified in order to match the registration time of the conventional weather stations, after which the daily $T_{min}$ and $T_{max}$ values were selected and compared with those obtained from MexHiResClimDB, Daymet and L15, as shown in Figure 15(a) and (b) for September 2013 and July 2010 respectively, along with the daily MAE determined for each dataset.

The daily variation of both $T_{min}$ and $T_{max}$ is, in general, well represented by all gridded datasets and the largest MAE was obtained at the AWS 806 for the two months analysed: for September 2013 Daymet had a MAE = 7.7 and 7.1 °C for $T_{min}$ and $T_{max}$, while L15 had a MAE of 9.1 and 4.8 °C for the considered temperatures. Finally, the plots for AWS 810 – located at the Mexico-U.S. border, in the city of *Ojinaga* (Fig. 9) – show that the three datsets provide similar temperature values at that location, although the data from the MexHiResClimDB provided the lowest $T_{max}$ MAE for September 2013 and for $T_{min}$ of

July 2010 (1.1 and 1.6 °C respectively).

To summarize the results of the 32 comparisons shown in Figure 15(a) and (b), for both $T_{min}$ and $T_{max}$, the MAE obtained with the MexHiResCimDB was the lowest in 15 and 14 cases ($T_{min}$ and $T_{max}$ respectively) followed by Daymet (11 and 14 cases), while L15 provided the lowest MAE for 6 and 4 cases.

The daily differences between the gridded temperature datasets and the AWS data are summarized in Fig. 16, where it can

be seen that temperature data from MexHiResClimDB exhibit the lower MAE of all datasets (1.7 and 1.8 °C for $T_{min}$ and $T_{max}$ respectively), followed by Daymet (2.0 °C for both temperatures) and L15 (2.4 and 2.5 °C).

### 5.7  Climate summary and anomalies

The MexHiResClimDB (Carrera-Hernández, 2025a) provides daily, monthly and yearly data for $T_{min}$, $T_{max}$, Precip and also for $T_{avg}$ (Carrera-Hernández, 2025b, c, d, e, f, g, h, i, j) – which is a derived product of the interpolated temperature variables –

along with monthly and yearly climate normals of the four previously mentioned variables (Carrera-Hernández, 2025k, l, m, n, o).

**Table 3.** Country wide (a) Maximum and (b) Minimum values of daily, monthly and yearly averaged values for $T_{min}$, $T_{avg}$ and $T_{max}$.

(a) Maximum values

| $T_{min}$ | | | | | | $T_{avg}$ | | | | | | $T_{max}$ | | | | | |
|---|---|---|---|---|---|---|---|---|---|---|---|---|---|---|---|---|---|
| daily | | monthly | | yearly | | daily | | monthly | | yearly | | daily | | monthly | | yearly | |
| date | temp (°C) | date | temp (°C) | date | temp (°C) | date | temp (°C) | date | temp (°C) | date | temp (°C) | date | temp (°C) | date | temp (°C) | date | temp (°C) |
| 2020-08-31 | 19.7 | 2020-07 | 18.9 | 2020 | 13.8 | 2018-07-24 | 26.8 | 1998-06 | 25.8 | 2020 | 21.1 | 1998-06-15 | 35.4 | 1998-06 | 34.2 | 2020 | 29.2 |
| 2020-07-11 | 19.6 | 2020-08 | 18.8 | 2015 | 13.6 | 2017-06-23 | 26.6 | 2019-08 | 25.7 | 2017 | 21.0 | 2011-05-28 | 35.2 | 1980-06 | 33.6 | 2017 | 29.0 |
| 2020-07-09 | 19.4 | 2019-08 | 18.7 | 2019 | 13.6 | 2018-07-25 | 26.6 | 2020-07 | 25.5 | 2019 | 20.9 | 1951-06-19 | 35.2 | 2011-06 | 33.5 | 2011 | 28.8 |
| 2020-08-30 | 19.4 | 2016-07 | 18.6 | 2018 | 13.5 | 2020-07-13 | 26.6 | 2020-08 | 25.5 | 2016 | 20.8 | 2011-05-27 | 35.1 | 2003-05 | 33.4 | 2019 | 28.8 |
| 2020-07-08 | 19.4 | 1957-07 | 18.5 | 2017 | 13.5 | 2020-07-12 | 26.5 | 1980-06 | 25.4 | 2018 | 20.7 | 1951-06-18 | 35.1 | 2005-06 | 33.4 | 1998 | 28.8 |
| 2020-08-29 | 19.3 | 1960-07 | 18.5 | 2016 | 13.5 | 2020-07-11 | 26.5 | 1980-07 | 25.3 | 2015 | 20.6 | 2018-05-31 | 35.0 | 1960-06 | 33.2 | 1995 | 28.7 |
| 2020-08-13 | 19.3 | 1969-07 | 18.5 | 1958 | 13.3 | 1998-07-13 | 26.5 | 2016-07 | 25.2 | 1995 | 20.6 | 2003-05-17 | 35.0 | 1998-05 | 33.2 | 1996 | 28.7 |
| 2020-07-13 | 19.3 | 1980-07 | 18.4 | 2014 | 13.2 | 1951-06-19 | 26.5 | 1960-06 | 25.2 | 1994 | 20.6 | 1998-06-19 | 35.0 | 1996-05 | 33.2 | 1999 | 28.6 |
| 1998-07-13 | 19.3 | 1953-07 | 18.4 | 1957 | 13.2 | 1998-07-14 | 26.5 | 1998-07 | 25.2 | 1954 | 20.5 | 2003-05-18 | 34.9 | 1953-06 | 33.2 | 2009 | 28.6 |
| 1998-07-14 | 19.3 | 1998-07 | 18.4 | 1994 | 13.1 | 2018-07-23 | 26.4 | 1953-06 | 25.2 | 2014 | 20.5 | 2018-06-02 | 34.9 | 2019-08 | 33.1 | 1953 | 28.6 |

(b) Minimum values

| $T_{min}$ | | | | | | $T_{avg}$ | | | | | | $T_{max}$ | | | | | |
|---|---|---|---|---|---|---|---|---|---|---|---|---|---|---|---|---|---|
| daily | | monthly | | yearly | | daily | | monthly | | yearly | | daily | | monthly | | yearly | |
| date | temp (°C) | date | temp (°C) | date | temp (°C) | date | temp (°C) | date | temp (°C) | date | temp (°C) | date | temp (°C) | date | temp (°C) | date | temp (°C) |
| 1962-01-12 | 1.3 | 1987-01 | 5.7 | 1987 | 12.1 | 1962-01-11 | 9.0 | 1985-01 | 13.1 | 1976 | 19.4 | 1967-01-10 | 15.2 | 1992-01 | 20.0 | 1976 | 27.1 |
| 2011-02-04 | 1.8 | 1967-01 | 5.7 | 1975 | 12.1 | 2011-02-04 | 9.1 | 1966-01 | 13.2 | 1968 | 19.6 | 1967-01-09 | 16.0 | 1985-01 | 20.6 | 1966 | 27.2 |
| 1962-01-11 | 2.3 | 1951-01 | 5.9 | 2010 | 12.2 | 1967-01-10 | 9.2 | 1958-01 | 13.3 | 1966 | 19.6 | 1967-01-11 | 16.1 | 1966-01 | 20.6 | 1968 | 27.2 |
| 1951-02-03 | 2.5 | 1964-01 | 5.9 | 1999 | 12.2 | 1967-01-09 | 9.6 | 1964-01 | 13.4 | 1987 | 19.6 | 1962-01-11 | 16.1 | 1958-01 | 20.9 | 1992 | 27.4 |
| 1951-02-02 | 2.6 | 1973-12 | 5.9 | 1976 | 12.2 | 1962-01-12 | 9.7 | 1987-01 | 13.6 | 1975 | 19.7 | 1992-01-16 | 16.6 | 1981-01 | 21.1 | 1984 | 27.4 |
| 1997-12-15 | 2.6 | 1976-01 | 6.0 | 1970 | 12.3 | 1967-01-11 | 9.9 | 1973-01 | 13.7 | 1984 | 19.7 | 1992-01-17 | 16.7 | 2007-01 | 21.2 | 1958 | 27.5 |
| 1962-01-13 | 2.7 | 1999-12 | 6.0 | 1979 | 12.3 | 2011-02-03 | 9.9 | 1979-01 | 13.7 | 1973 | 19.7 | 1981-01-18 | 16.7 | 1976-12 | 21.3 | 1985 | 27.5 |
| 1973-12-21 | 2.7 | 2010-12 | 6.1 | 1973 | 12.3 | 1997-12-13 | 10.0 | 1967-01 | 13.8 | 2010 | 19.7 | 2011-02-04 | 16.8 | 1984-01 | 21.4 | 1987 | 27.6 |
| 1997-12-14 | 2.8 | 1960-02 | 6.1 | 1971 | 12.4 | 1971-01-07 | 10.3 | 1981-01 | 13.8 | 1964 | 19.8 | 2011-02-03 | 17.0 | 1964-01 | 21.4 | 1973 | 27.6 |
| 2011-02-05 | 2.9 | 1985-01 | 6.1 | 1974 | 12.4 | 1964-01-14 | 10.3 | 1992-01 | 13.8 | 1985 | 19.8 | 1985-01-13 | 17.1 | 1979-01 | 21.5 | 1964 | 27.6 |

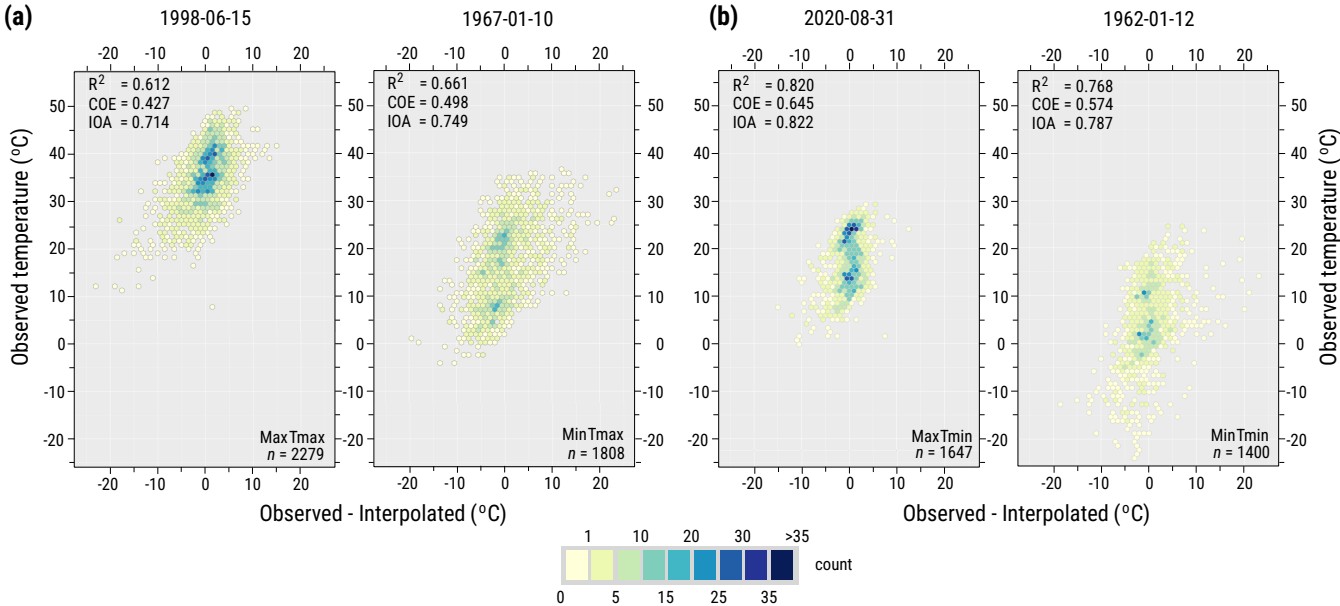

**Figure 14.** Scatterplots for minimum and maximum temperatures of (a) $T_{max}$ and (b) $T_{min}$ obtained through cross-validation.

This new database has been used to determine climate extremes (both minimum and maximum, as shown in Tables 2 and 3). To show how the climate variables vary through the 1951–2020 period according to the different aggregation times on which they are distributed, Fig. 17 was created, where it can be seen how temperature has increased in recent years.

It is interesting to note that for daily temperature, the distribution of warmer days for the three temperatures shown in Fig. 17 has an hourglass shape, thus showing colder temperatures for the 1961–1990 period – in particular for those years between 1970–1979. However, the bottom of this hourglass (for the three temperatures reported) is narrower than it is at its top, which clearly shows an increase in temperature – which is easier to note when $T_{min}$ (Fig. 17(c)) is observed, as the days with $T_{min}$ between 17.5–20.0 °C have increased since the summer of 2013. The increase in daily temperature observed in this Figure is consistent with the values of Table 3, where it can be seen that the eight days with the highest $T_{min}$ occurred in 2020. On a monthly basis, this warming can also be seen in Fig. 17 because 2019 was the year where three months (Jun-Aug) exceeded 32.5°C, while on 2020 four months (May-Aug) also exceeded this temperature (Fig. 17(a)); similar warmer blocks are also observable for both $T_{avg}$ and $T_{min}$. By looking at the five-year moving average of monthly temperature, it can be seen that $T_{min}$ started to increase in 2013 and by 2020 most months show $T_{min}$ anomalies above 1°C, with July showing the largest increase for both $T_{min}$ and $T_{max}$.

The summary plots of precipitation (Fig. 17(d)) clearly show that the rainy season in Mexico is from May to October, with the wettest months being July, August and September. The monthly plot of precipitation clearly shows that January of 1992 was an exceptionally wet January, while the yearly plot shows that 1958 was the wettest year for the period covered by the MexHiResClimDB. However, although the aforementioned extremes can be easily seen on Fig. 17(d), an increase or decrease

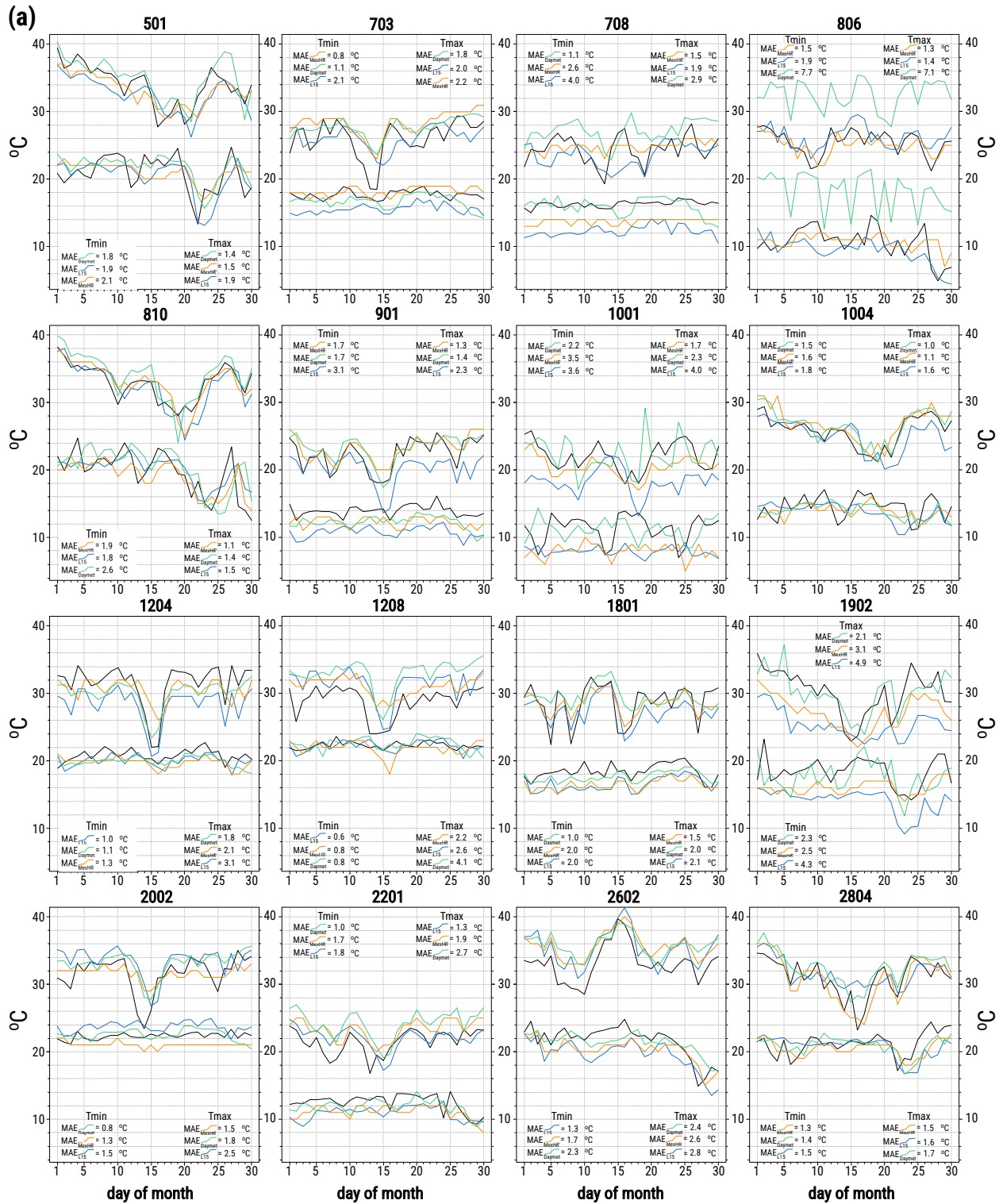

**Figure 15.** Comparison of daily temperature obtained with data from Weather Stations, MexHiResClimDB, Daymet and L15 for (a) September 2013, and (b) July 2010. This figure shows the limitation of storing temperature data in the MexHiResClimDB as integers ($T_{min}$) for AWSs 708 and 2002.

**(b)**

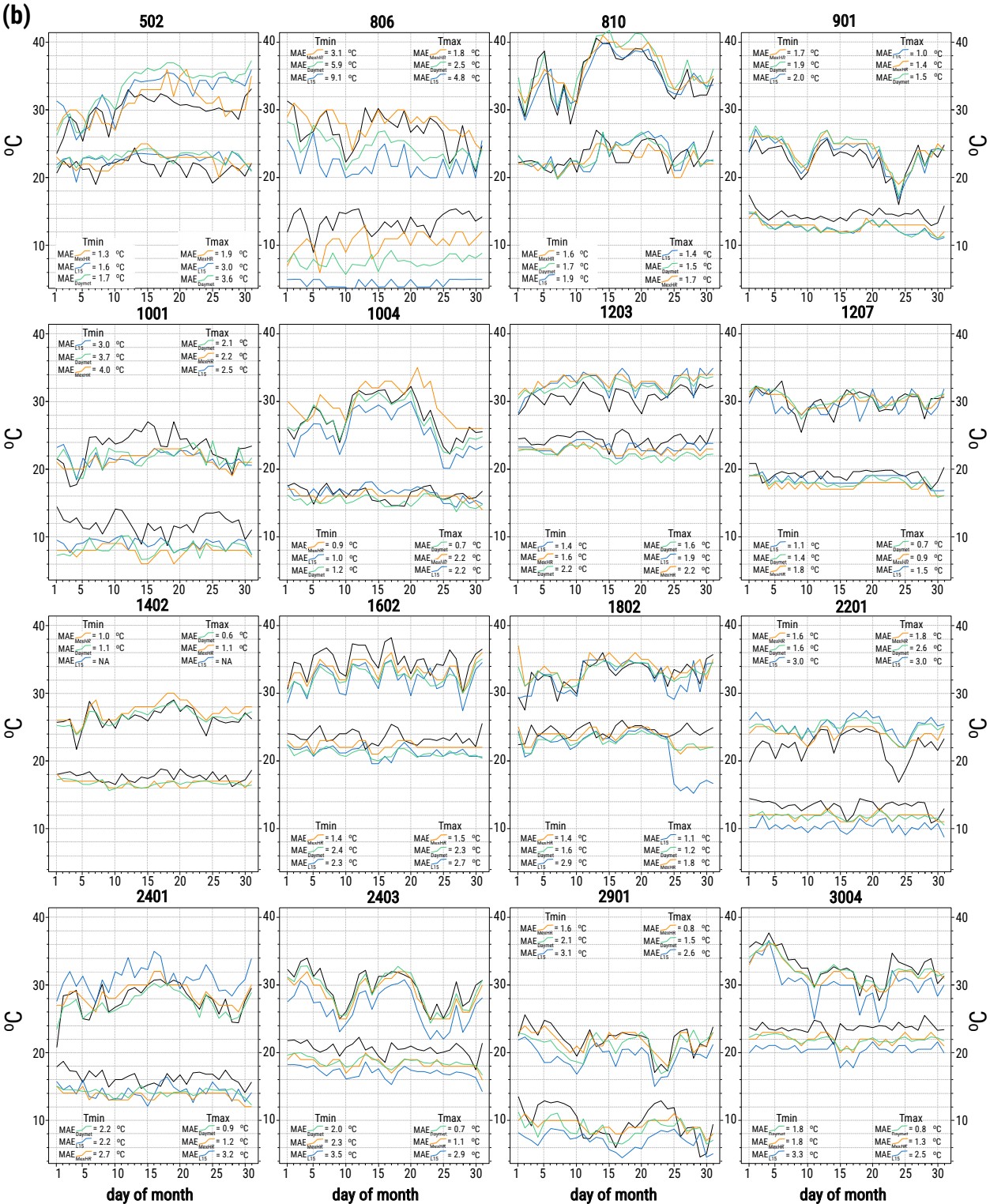

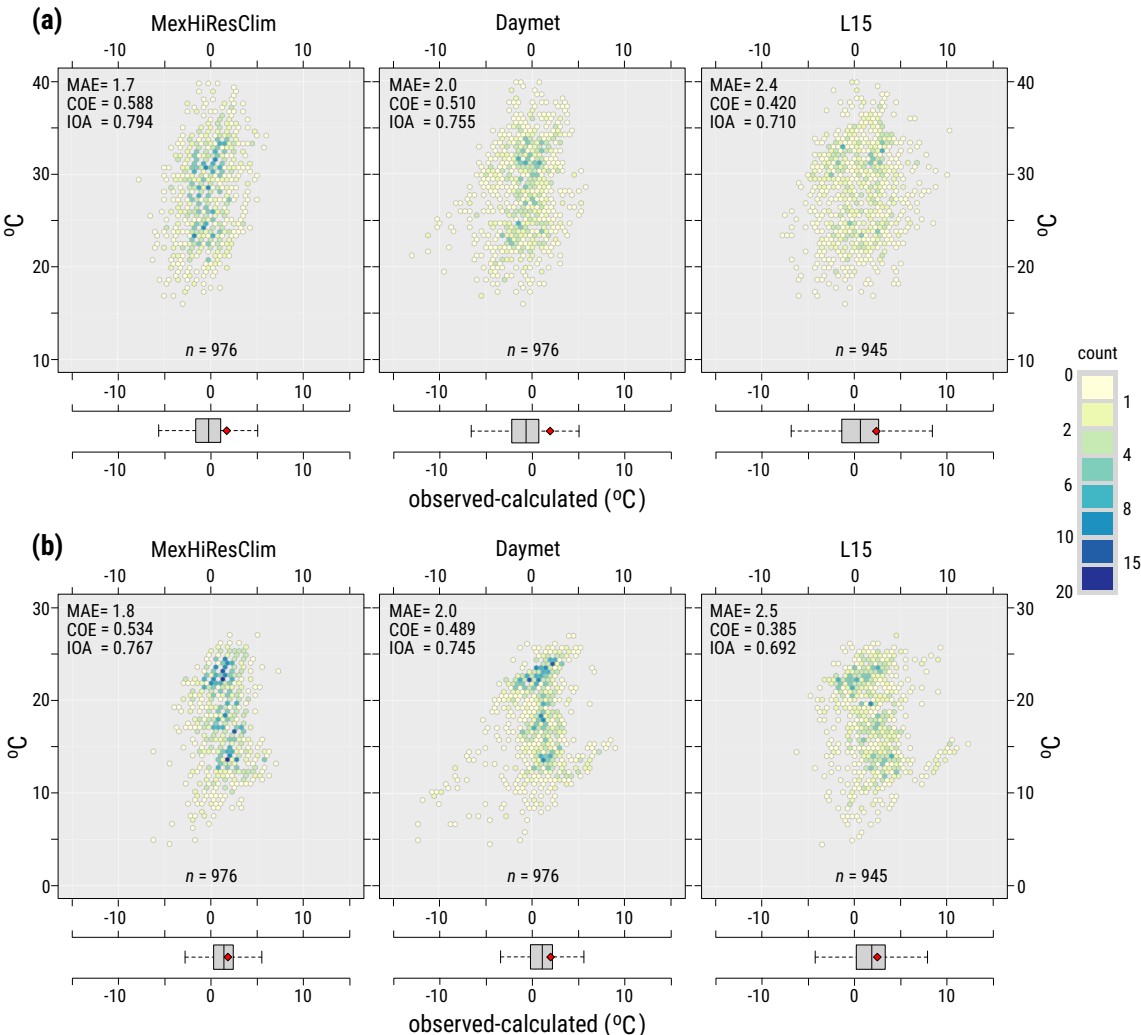

**Figure 16.** Validation of gridded temperature using data from the Automatic Weather Stations and 61 days sampled at 32 locations for (a) Tmax and (b) Tmin. It should be noted that the L15 dataset has a data gap on the western region of Central Mexico, which is why both the $T_{min}$ and $T_{max}$ hex-bin scattergrams were developed using 945 points instead of the 976 points used for MexHiResClimDB and Daymet.

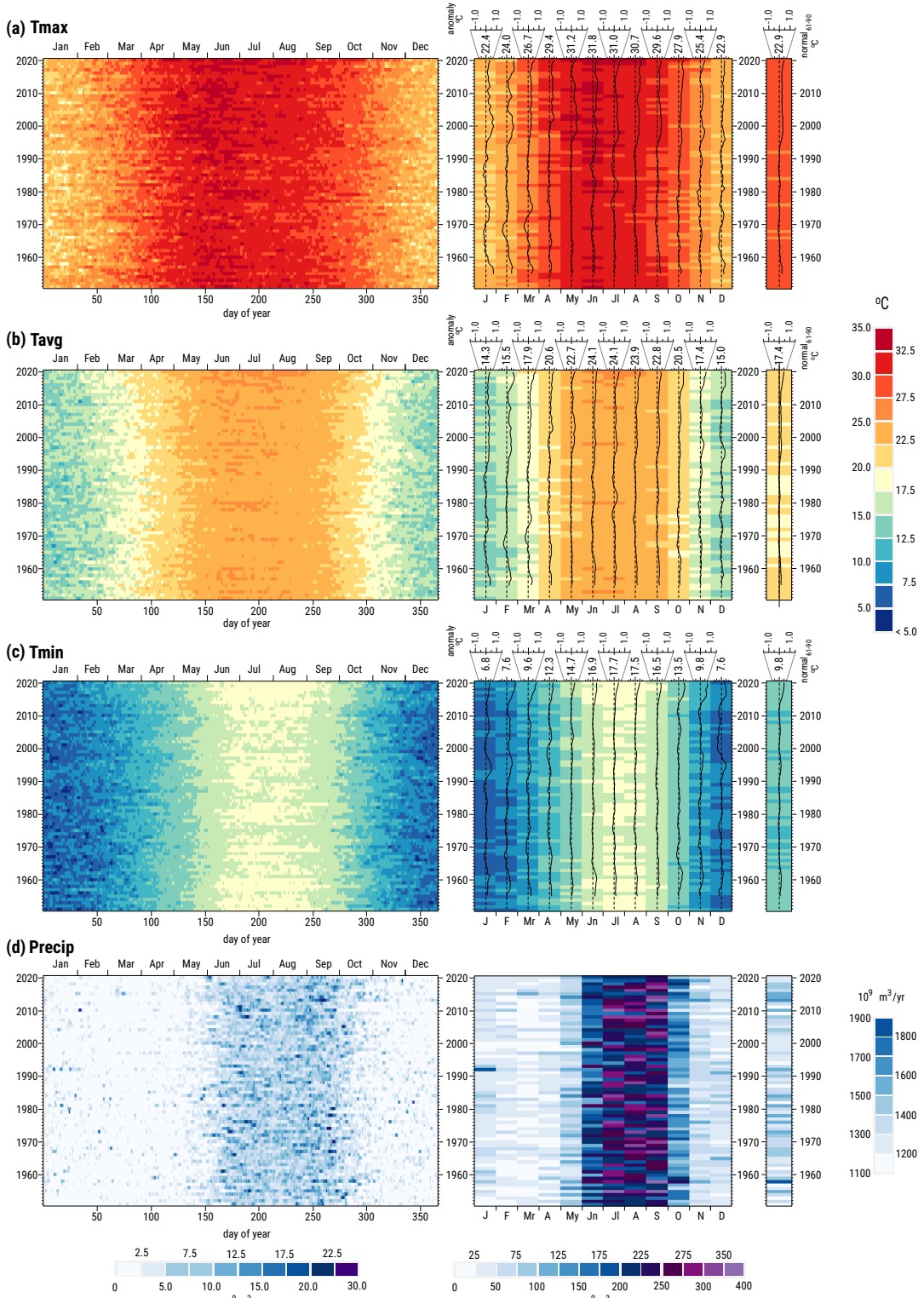

**Figure 17.** Countrywide daily, monthly and yearly values for the climate variables derived in this work. Precipitation values are given in volume ($10^9$ m$^3$) and it can clearly be seen that Mexico's rainy season is from May to October, that 1958 was the year with the largest precipitation for the 1951–2020 period, and that January 1992 was an exceptionally wet month. The variability of the five-year moving average of temperature with respect to the 1961-1990 normal is also shown.

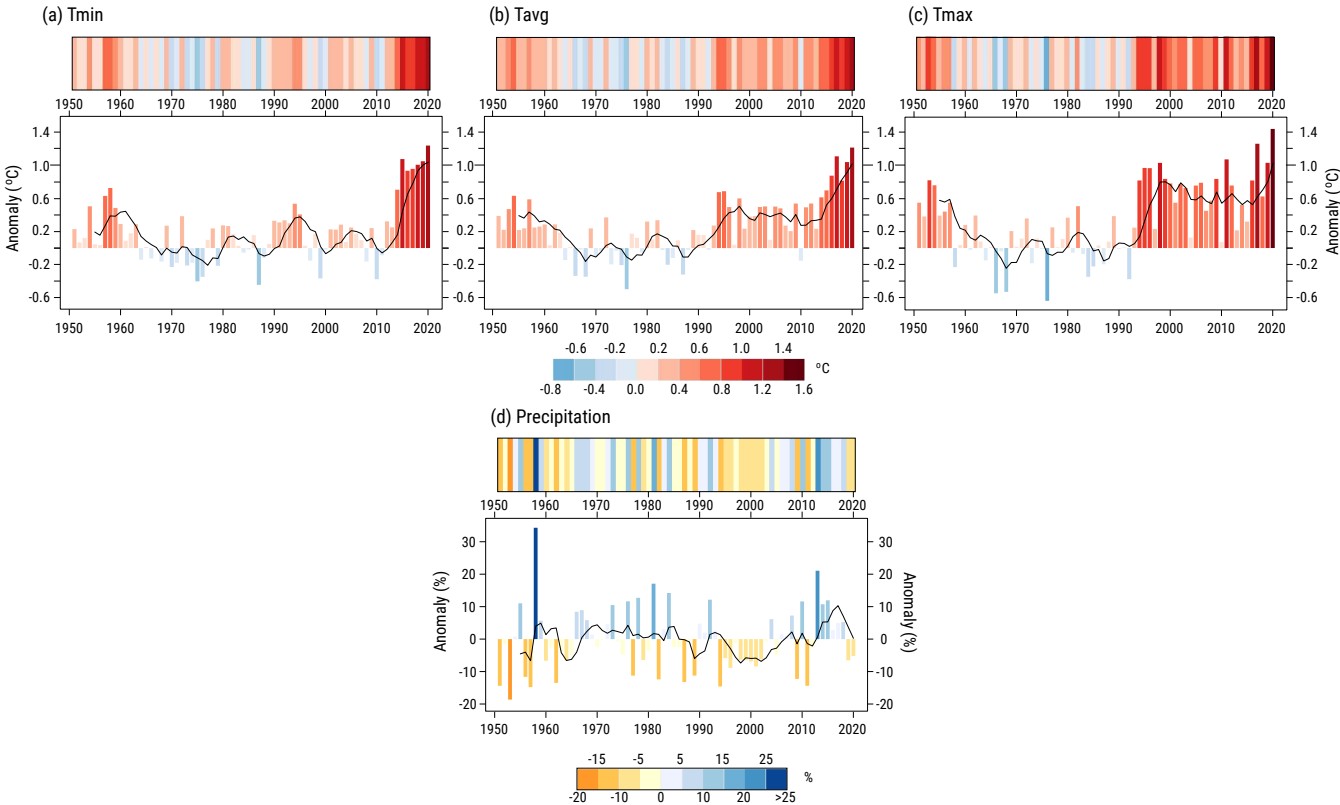

**Figure 18.** Yearly anomalies with respect to the 1961–1990 normals of (a) $T_{min}$, (b) $T_{avg}$, (c) $T_{max}$ and (d) Precipitation

in precipitation is difficult to observe, as is the case for yearly temperatures (due to the color scale selected in order to represent
the temperature range between $T_{min}$ and $T_{max}$). To provide further insight into this situation, the yearly anomalies of the four
climate variables available on the MexHiResClimDB were computed for the 1951–2020 period as shown in Fig. 18 in both
the well known "warming stripes" format and as bar plots (considering 1961–1990 as the baseline period, according to the
recommendation of the World Meteorological Organization, (WMO, 2017)).

The temperature anomalies shown in Fig. 18 show a warm period before 1964, and a fluctuation between colder and warmer
periods between 1964–1990, although the anomalies for $T_{min}$ (Fig. 18(a)) still fluctuated until 2010. However, in contrast to
Fig. 17, the anomalies shown in Fig. 18 clearly increase, in particular for $T_{max}$, because its positive anomalies (which started in
1992) reached 1.0 °C above the 1961–1990 normal in 1998 (thus yielding a $T_{max}$ increase rate of 0.13 °C/yr for the 1992–1998
period) and an anomaly of 1.4 °C in 2020, with a five-year moving average of around 0.6 °C between 2000–2016 that reached
1.0 °C in 2020 (Fig. 18(c)). For the 2016–2020 period the $T_{max}$ increase rate was similar to that of 1992.

As previously mentioned, the $T_{min}$ anomalies fluctuated between cold and warm periods until 2012, when the $T_{min}$ anomaly
started to increase from two cold years (2010 and 2011) to 0.2 °C in 2012 and to 1.2 °C in 2020 (Fig. 18(a)), which represents

a warming rate of 0.15 °C per year. Finally, the $T_{avg}$ anomalies show a warming tendency that started in 1990 that drastically increased in 2014, with a maximum value of 1.2 °C in 2020.

The precipitation anomalies shift between dry and wet spells, although the 1995–2003 were dry years which were folowed by a couple of wet years before the 2009 and 2011 dry years (in fact, the five-year moving average shows a dry spell of 12 years between 1995–2007). This tendency in precipitation is important for water management and water supply, because even though the 2013–2020 five-year moving average shows a seven year wet period, 2019 and 2020 were dry years. When this tendency is compared with that of temperature, it can be concluded that higher temperatures will increase evapotranspiration and less water will be available. However, it should be kept in mind that even though the plots of Fig. 18 show that Mexico is warming, further studies are needed in order to pinpoint the areas where climate change is having a profound impact – for example, at the watershed level – a task that can be now done using the MexHiResClimDB.

## 6   Conclusions

This work presents Mexico's High Resolution Climate Database (MexHiResCLimDB), which is a new gridded, high-resolution ($\approx$600 m) climate dataset comprised of daily, monthly and yearly precipitation and temperature ($T_{min}$, $T_{max}$, $T_{avg}$). The monthly and yearly values were derived from daily interpolations obtained by using Kriging with External Drift on a local neighbourhood through `gstat` within `R` and `GRASS` along a relational database developed in `PostgreSQL`.

Although different gridded climate datasets that cover Mexico are currently available, the MexHiResClimDB improves the spatio-temporal representation of climate variables over the country, as it is now the gridded climate database with the largest time coverage (1951–2020) and the highest spatial resolution (with 20" or $\approx$ 600 m). Furthermore, the daily precipitation data provided by this database adequately represents the spatial variation of the extreme precipitation events that occurred during September 15–16 of 2013, caused by the presence of Tropical storm Manuel in the Pacific Ocean and Hurricane Ingrid (Cat 1) in the Gulf of Mexico. Using data from 61 days retrieved from Automated Weather Stations located throughout Mexico, it was found that precipitation data from MexHiResClimDB has a MAE=8.7 mm, followed by L15 (9.5 mm), Daymet (10.1 mm) and CHIRPS (11.7 mm). For $T_{min}$ and $T_{max}$, the lowest MAE was obtained with MexHiResClimDB (1.7°C and 1.8 °C, respectively), followed by Daymet (2.0 °C for both temperatures) and L15 (2.4°C and 2.5 °C).

With this new database it was possible to summarize extreme events of precipitation and temperature in Mexico for the 1951–2020 period – a summary that was not available before. With this summary, it was found that the wettest year was 1958, the wettest day 1970-09-26 (which surprisingly was not caused by a hurricane) and that September of 1974 had the largest precipitation events for two and three consecutive days (21–22 and 20–22 respectively) – with September of 2013 being the wettest month. Regarding temperature extremes, it was found that eight out of the ten days with the highest $T_{min}$ occurred in 2020, the two months with the highest $T_{min}$ were July and August of 2020 and that the six years with the highest $T_{min}$ were 2015–2020. When $T_{max}$ was analysed, it was found that the hottest day was 1998-06-15, while June of 1998 was the hottest month and 2020 the hottest year, and that the four hottest years occurred between 2011–2020.

**Table 4.** Datasets included in Mexico's High Resolution Climate Database (MexHiResClimDB, Carrera-Hernández (2025a)). The daily datasets are comprised of data for all days of the 1951–2020 period, monthly data for all months (i.e., 12 months × 70 years) and yearly data for all years (70).

| Citation | Data | Digital Object Identifier |
|---|---|---|
| Carrera-Hernández (2025b) | Daily $T_{min}$ for 1951–2020 | DOI:10.6084/m9.figshare.28462808 |
| Carrera-Hernández (2025c) | Daily $T_{avg}$ for 1951–2020 | DOI:10.6084/m9.figshare.28462835 |
| Carrera-Hernández (2025d) | Daily $T_{max}$ for 1951–2020 | DOI:10.6084/m9.figshare.28462820 |
| Carrera-Hernández (2025e) | Daily Precip for 1951–2020 | DOI:10.6084/m9.figshare.28462796 |
| Carrera-Hernández (2025f) | Monthly $T_{min}$ for 1951–2020 | DOI:10.6084/m9.figshare.28124789 |
| Carrera-Hernández (2025g) | Monthly $T_{avg}$ for 1951–2020 | DOI:10.6084/m9.figshare.28462769 |
| Carrera-Hernández (2025h) | Monthly $T_{max}$ for 1951–2020 | DOI:10.6084/m9.figshare.28462679 |
| Carrera-Hernández (2025i) | Monthly Precip for 1951–2020 | DOI:10.6084/m9.figshare.28462787 |
| Carrera-Hernández (2025j) | Yearly data for $T_{min}$, $T_{avg}$, $T_{max}$ and Precip. | DOI: 10.6084/m9.figshare.28074998 |
| Carrera-Hernández (2025k) | Monthly and yearly normals (1951–1980) for $T_{min}$, $T_{avg}$, $T_{max}$ and Precip. | DOI: 10.6084/m9.figshare.28464398 |
| Carrera-Hernández (2025l) | Monthly and yearly normals (1961–1990) for $T_{min}$, $T_{avg}$, $T_{max}$ and Precip. | DOI: 10.6084/m9.figshare.28464458 |
| Carrera-Hernández (2025m) | Monthly and yearly normals (1971–2000) for $T_{min}$, $T_{avg}$, $T_{max}$ and Precip. | DOI: 10.6084/m9.figshare.28464461 |
| Carrera-Hernández (2025n) | Monthly and yearly normals (1981–2010) for $T_{min}$, $T_{avg}$, $T_{max}$ and Precip. | DOI: 10.6084/m9.figshare.28464488 |
| Carrera-Hernández (2025o) | Monthly and yearly normals (1991–2020) for $T_{min}$, $T_{avg}$, $T_{max}$ and Precip. | DOI: 10.6084/m9.figshare.28464494 |

The anomalies obtained with this dataset show an undeniable increase of temperature in Mexico; however, further studies are needed in order to pinpoint the areas where climate change is having a profound impact – a task that can be now done using the MexHiResClimDB due to its spatio-temporal resolution.

## 7 Data availability

The datasets included in Mexico's High Resolution Climate Database (MexHiResClimDB, Carrera-Hernández (2025a)) are distributed as GeoTiffs (and not in NetCDF format in order to provide data for the 366 days of leap years); in addition to Precip, $T_{min}$, and $T_{max}$, the derived $T_{avg}$ from the aforementioned temperature values is also distributed. The available data are distributed at different aggregation (precip) or average (temperature) time steps (daily, monthly and yearly) as well as monthly and yearly normals for five different periods (1951–1980, 1961–1990, 1971–2000, 1981–2010 and 1991–2020). The datasets are provided on geographic coordinates referred to the WGS84 ellipsoid and are available on Figshare through the links provided in Table 4.

## 8 Code availability

A code snippet showing how Kriging can be parallelized in R using data stored in both PostgreSQL and the GRASS GIS is provided in Carrera-Hernández (2025p).

**Competing interests**

The author declares no conflicts of interest.

**Acknowledgements**

Funding for this project was provided by UNAM through project grant PAPIIT-IN101524. ALOS AW3D30 Digital Elevation Model provided by JAXA. The processing and generation of data, as well as the figures and this manuscript were all created with Open Source Software (PostgreSQL, R, GRASS GIS, QGIS, Inkscape and LaTeX) on Linux.

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
