# Peer review of "Mexico's High Resolution Climate Database (MexHiResClimDB): a new daily high-resolution gridded climate dataset for Mexico covering 1951–2020"

_Earth System Science Data, 2025_

## Author Comment (AC2)

I thank R1 for the thorough review and the time taken to review this manuscript; the comments and suggestions provided will improve the manuscript.

**My answers to R1's suggestions/observations are written in blue.**

General comments: The author presents a newly developed gridded, high-resolution climate dataset comprised of daily, monthly and yearly precipitation and temperature for Mexico that they have developed using stations data and Kriging with External Drift on a local neighborhood (KEDI) interpolation. The study presents a new dataset that can be very useful in understanding different aspects of climate change and its impacts at relatively high spatial and temporal resolution across Mexico.

This dataset appears to be a major improvement over existing datasets for Mexico. The station counts for the 1999-2020 period are much higher than in CRU and GPCC datasets; it would be helpful to demonstrate that they are higher than the Daymet dataset. We conducted a visual comparison of the MexiHiRes January and July 1981-2020 normals to the WorldClim, CHELSA, Daymet, and US PRISM products, and found that the MexiHiRes surfaces were relatively free of artefacts and appeared to be more credible than WorldClim, CHELSA, and Daymet, and generally more compatible with the adjacent US PRISM normals.

**I will include this analysis in the new version of the manuscript; thank you for suggesting it.**

Our main concerns relate to the filtering of the original datasets—specifically, the criteria used, the quality of the input data, and the details of the interpolation method. The data and methods section needs substantial revision. Clear explanations, potentially a flowchart, and a discussion of how each step in the process might influence the results would be highly beneficial. The validation and comparison presented are limited and only applied to specific extreme cases. The manuscript lacks discussion of potential limitations of the datasets. It is not sufficiently demonstrated that the newly generated datasets are better than existing alternatives, despite multiple claims by the author, though we are confident this can be done with more convincing analyses.

Although this paper requires major revisions, it appears that this is an exciting and necessary improvement to climate data available for Mexico, and we commend the author for the effort.

Thank you for your positive comments; I will address your suggestions and observations in the revised manuscript according to my answers that follow your following comments:

**Major comments:**

• Description of method: The author mentions in multiple instances that this is better data. However, the methods is not clear on the process of interpolation of stations data. How the observation data, how the interpolation works, what are benefits or

limitations of the method they choose and why they chose the method need to be clearly explained.

• Validation and comparison: The intercomparison with other datasets wasn't convincing. The author needs to make a more comprehensive comparison of their new dataset to other datasets, not only for selected extreme events. Further, they have to present how their data represents climate differently than other datasets and why it is so. I also suggest that the author discusses their methodological and data limitations.

**I agree; I will include a section on the limitations of this new database.**

• Climate normals: The focus of this paper is on the daily time series. However, the climate normals are also an important contribution and deserve some description, validation, and intercomparison.

Thanks for this suggestion; I think than this will improve the comparison section and will include it in the revised version of the manuscript.

• Content Refinement: The Introduction is ambiguous and there are several instances when concise refinement of the writing is required. They have to present the need for this work, and how this dataset will be better than existing datasets. Overall writing can be more concise.

**Other comments:**

In the comments below. L indicate line number, \* indicate major comments.

• L3: remove terms like largest , highest etc and provide specific values like what temporal coverage, especially in abstract.

**I agree and will remove these adjectives where possible.**

• \*L10-17: not clear if the author is just presenting these values or saying that these values are more realistic than values from other datasets. I am not clear what the author is trying to say by saying 'a summary that was not available before'. It is available from other datasets like ERA5land. Reliability of the values can be different, but it is available.

I am presenting these values because to date there is no information anywhere regarding country-wide information on climate extremes for Mexico's. The information could be extracted from ERA51and, but to my knowledge no one has summarized this info before.

**L12-17: for which period are these values true, 1951-2020? specify.**

Yes, those values are for the 1951-2020 period; I did not want to mention again "for the 1951-2020 period...", because I thought it would be too repetitive. I can rewrite those lines as follows:

With this new database it was possible to summarize extreme events of precipitation and temperature in Mexico for the 1951-2020 period - a summary that was not available before: the wettest year was 1958, the wettest day 1970-09-26, and September of 2013 the wettest month. Regarding temperature, it was also found that for the 1951-2020 period, eight out of the ten days with the highest  $T_{min}$ occurred in 2020, the two months with the highest  $T_{min}$  were July and August of 2020 and that the six years with the highest  $T_{min}$  were 2015-2020. When  $T_{max}$  was analyzed, it was found that the hottest day was 1998-06-15, while June of 1998 was the hottest 15 month and 2020 the hottest year, and that the four hottest years occurred between 2011-2020. Nationwide (and considering 1961-1990 as the baseline period),  $T_{min}$ ,  $T_{avg}$  and  $T_{max}$  have increased, with their anomalies drastically increasing in recent years and reaching values above 1.0 °C in 2020.

• L18-19: generic

I wanted to emphasize that the spatial resolution of this new database allows the development of hydrological studies for small watersheds.

• L39: define for the first use

I guess that R1 refers to MODIS; I will describe it in the new version.

- L46-48: redundant
- L46 -48: superfluous. This whole paragraph can be summarized to few sentences, much of the details may not be relevant to the paper

I agree; I can cite the recent review of Mankin et al.: Review of gridded climate products and their use in hydrological analyses reveals overlaps, gaps, and the need for a more objective approach to selecting model forcing datasets (2025), Hydrol. Earth Syst. Sci., 29,85-108, https://doi.org/10.5194/hess-29-85-2025

• \*L63 (whole paragraph): I suggest that the author create a table with name of dataset, resolution ( spatial , temporal), data period, region and citation . such table will give this exact information from this and previous paragraphs and will be more succinct

Thank you for this suggestion; I will create a table with this information.

• L75: "monthly surfaces of precipitation, Tmin and Tmax for the 1910–2009 period (i.e. 12 surfaces in total)". If these were normals, wouldn't this be 36 surfaces? But also, the term "monthly surfaces" is ambiguous about whether it is a monthly gridded time series or climatological average/normal. As suggested, and table would be more succinct and explicit.

I agree; these surfaces are not normals; they are monthly averages for the 1910-2009 period. I also agree that this information is better shown on a table.

• \*L80: what about Daymet / ERA5 or other regional and global datasets that covers Mexico and available at daily resolution but may not be as same spatial resolution as this dataset ? However, What problem their spatial resolution creates that this MexHiresClimDB will better address ? Need to explain why this work is important and what value it adds in more detail.

I will create a table and move lines 200-239 to this section (as suggested by R1 in another comment).

• L93: use numeric for XXth

Thank you for noticing this; I have already modified it.

• L95: what is the link between the description of study area and the datasets the author developed ?

Thank you for pointing this out; I have already modified the manuscript and it now reads as follows:

Due to Mexico's geographic context, its precipitation is extremely variable, as on a given day during the rainy season, precipitation can vary from 0 to more than 300 millimetres and the adequate representation of these extreme events is important in Mexico, in particular for those cities that depend on reservoirs that are only filled during hurricanes or where flash-floods are a recurrent event.

- L97-99: move this paragraph after the data description
- L100: what aforementioned refer to ? Provide the name of dataset. Throughout the paper try to use 'aforementioned' less often as it creates confusion

Thanks for the suggestion; this "aforementioned" refers to the database that I developed in PostgreSQL. These lines will be improved with the flow chart that I will include in the new version of the manuscript.

• L100-101 : rewrite: how many stations originally were there , how many were used ? what are the criteria to select the stations

The total number of stations with data is 5467; I will add this to the manuscript. The current version of the manuscript reads (L101-103) "... and once the climate records were in PostgreSQL only those stations with more than 10 years of registered data were selected; accordingly, not all available stations were used, and the number of stations varied across the 1951-2021 period, as shown in Fig. 2."

Also, the caption of Fig. 2 reads "Number of weather stations used for daily interpolations of (a) temperature, (b) precipitation".

• L108: what might have caused less number of stations after 2021 ?

I think that this was caused by a reduction in budget and the believe that in situ measurements are not needed.

• \*Figure 2. The station coverage in this dataset is much better than in the CRUts and GPCC datasets, which report a dramatic (>90%) decline in station coverage after 1998. This is a compelling advantage of the MexiHiRes product and it would be useful to highlight it.

**Thank you for your suggestion; I will highlight it.**

• \*L110 onwards to paragraph: need more description of the process and how this method works. For example, what happens to temperature lapse rate and what value is used ? A flowchart with all the steps involved from data collection to final output will be useful.

Kriging with External Drift considers the relationship between temperature and elevation, because elevation is used as an auxiliary variable. I think that the suggestion of including a flowchart describing the procedure will improve the manuscript, and I will include it in the revision.

**\*L125-128: can the author show how sensitive their used values are ?**

I will describe this in more detail and will rewrite these lines. The revised manuscript will include the following:

"These values were recommended by Carrera-Hernandez et al., (2024) after a detailed comparison of different Kriging variants at both national and stratified domains showed that KED1 using elevation as a secondary variable provided the best representation of yearly precipitation in Mexico. The manuscript that details these analyses is currently under review (Hydrological Sciences Journal).

**• L129-133: can be more succinct**

I think that it is important to show the amount of time and resources required to develop this database, but will try to rephrase it.

• L134: recommend re-expressing  $1.227 \times 10^6$  as 28 months so readers don't have to do the math. It's a lot of computation!

Thanks for the suggestion; I agree with it.

• L129: Is 26GB minimum requirement? not clear

**Each interpolation takes 26GB of RAM, but I will rephrase it.**

• L146: "if R2=0.80, then the model explains 80% of the variability". Reword: it explains 80% of variance, not variability. since variance is calculated from squared deviations, this interpretation will overestimate the amount of variability/dispersion that is explained

Thanks for your suggestion; I will reword this line.

• L170: is it value or variable, be specific.

Thanks for this observation; it is variables and I have already corrected it.

• L172: it's true that MAE for raw precipitation (in mm) is meaningless, since an error of 50mm has a different significance in a wet vs dry climate. however, MAE is meaningful if precipitation is log-transformed prior to analysis of error. By the same token, I would expect that the other metrics (R-square, COE, and IOA) are confounded by raw precipitation values in mm; wouldn't they primarily represent wetter regions—and wet anomalies—where absolute errors (in mm) are larger.

Thanks for the comment; I will look into this and use a log-transformation for precipitation.

• \*L200-239: I suggest the author move the description of these datasets to introduction and summarize there. Here they directly present the results of comparison.

Thanks for your suggestion; I will move these lines to the introduction.

• \*L290-291: This is not well justified statement but based on selected extreme event and a single coefficient. How this statement compares with what they presented in figure 5. I recommend performing relative Root mean Square Error to explore relative goodness of each dataset for both temperature and precipitation

On lines 281-291 I compare both the COE and the IOA. I use the extreme events of Figure 6 because I believe that these extremes need to be well represented in gridded databases. I do not agree with R1's suggestion of using the RMSE in addition to the metrics shown in the manuscript (R2, COE and IOA), due to the reasons given on lines 160-165:

L160-163: The Mean Absolute Error (MAE) is also used in this work because it is an unambiguous and more natural measure of average error than the Root Mean Square Error (RMSE, Willmott and Matsuura (2005)) due to the bias of RMSE when large outliers are present (Legates and McCabe, 1999).

L164: The modified Index of Agreement (IOA, Legates and McCabe (1999)) has the advantage that errors and differences are not inflated by their squared values...

L167-168: Another advantage of the IOA is that it is related to the Mean Average Error (MAE) and the Mean Absolute Deviation (MAD)

• Fig 7. The panels are labelled a,b,c,b,c.

Thank you for pointing this error; I have corrected it.

• Fig 7. It is very hard to see the distributions at typical printing size. Recommend altering the color scheme to better stand out against the background.

Thanks for your suggestion; I will modify the background.

• L311: not clear what the author is trying to say here.

Thank you for pointing this out. I have corrected the manuscript. It now reads:

The MexHiResClimDB also includes temperature data, and the analysis of extreme values of temperature is useful in climate change analysis; however, there is currently no information available on the coldest or hottest days in Mexico for the 1951--2020 period.

• \*L325-326: similar comments for precipitation and temperature. I suggest the author adds average comparison as well as relative comparison among the datasets to provide better perception on how their new dataset is better than existing datasets, if it is.

Thanks for this suggestion; I plan to compare the normals from L15, daymet and MexHiResClimDB to improve these comparisons. Furthermore, I will use data from some of the automatic stations managed by Mexico's Meteorological Agency.

• L328: no need to write the full form again, be consistent

I decided to write the full form again to avoid monotony to the reader; I changed it to IOA in the new version of the manuscript.

• L330: "it is not possible to obtain cross-validation values for L15 or Daymet". This is a reason to use some other metric for intercomparison.

I will use data from automatic stations to improve the validation of the MexHiResClimDB

• L335: "neither L15 nor Daymet were capable of howing the temperature extremes that were obtained through the MexHiResClimDB". This is not apparent from figure 9. The maximum values seem similar to L15

I modified this line and it now reads as follows:

... a visual comparison of the spatial distribution of both  $T_{min}$  and  $T_{max}$  for the dates with minimum and maximum values of the aforementioned temperature values is done in Fig. 9, where it can be seen that the MexHiResClimDB is the database with the longest temporal coverage and that neither L15 nor Daymet were capable of showing the temperature extremes that were obtained through the MexHiResClimDB for the period covered by this new database.

• Figure 9: it is not clear what the author is trying to illustrate from these plots. It is already clear that their data has longer temporal coverage across Mexico than other mentioned datasets. It does not provide information on how better their dataset is compared to others.

One point on which this dataset is better than others is the temporal coverage and I decided to create this figure in order to highlight this point. This comment is linked to the previous point, and I think that it is important to highlight that neither L15 nor Daymet can be used to obtain the temperature extremes shown on Table 3 due to their temporal coverage.

• Figure 11: describe what climate stripes bars represent and what additional information they provide other than the anomaly plots. Otherwise remove

I think that showing the anomaly plots is a useful application of this new database; in addition, these plots also show the five year moving average of all variables and show the nation-wide warm up that is occurring in Mexico.

**• L393-L395: this has only been demonstrated for a few specific events, so the statement should likely be qualified as such.**

I agree and modified the lines accordingly to the following:

... the daily precipitation data provided by this database is the only one that adequately represents the spatial variation of the extreme precipitation events that occurred during September 15--16 of 2013, caused by the presence of Tropical storm Manuel in the Pacific Ocean and Hurricane Ingrid (Cat 1) in the Gulf of Mexico.

---

## Author Comment (AC3)

The author develops a complete daily gridded dataset of precipitation and temperature for Mexico at a very high resolution considering the extension of the spatial domain. The research is mostly correct; however, I have some concerns about the method used to develop the daily grid and the validation process.

I thank R2 for these positive comments and for the time taken to review this manuscript.

My answers to R2's suggestions/observations are written in blue.

The choice of an interpolation method is not easy, especially for precipitation, since it can yield very different results depending on the method and the parameters. The KED is a trustworthy option when used for a single timestep, for example to show the daily precipitation/temperature in one day/event (as shown in the examples of extreme events). However, when you apply the same method for a long-term climate time series without further corrections, you are introducing temporal biases that can lead to unwanted inhomogeneities both in temporal and spatial dimensions. This is not new and it is a basic assumption in gridded datasets creation as shown in the wide (not cited) scientific literature (e.g. https://doi.org/10.1002/wat2.1555, https://doi.org/10.1002/joc.1322,https://doi.org/10.1029/2008JD010100, https://doi.org/10.1559/152304085783914686 ). I recommend the author to make a deeper review on the requirements for creating reliable grids, starting by some of the datasets that are cited in the article. The actual problem with creating a single grid for each day, independently from the previous and following, is that the number and location of neighboring observations change with time, and that lead to biases that have a significant impact in, for example the analysis of trends or even in the aggregation at coarser temporal scales. In this regard, my second main concern is the validation approach.

Thanks for the recommendation; I will improve this section using the provided references.

It is ok to check the errors by series with the proposed statistical tests, however, you are comparing the complete series of observations with their corresponding predictions without considering the number of missing values, the differences by elevation ranges, by months, seasons, etc. It is difficult to see biases that are useful to interpret how reliable are the results. In addition, the kriging process usually provides a variance dataset, measuring the uncertainty associated with the prediction at every specific location. It would be useful to see an associated gridded dataset with the error/uncertainty for each day and variable, as done in many other datasets, to account for the reliability of each prediction and let the user decide how to use the information.

I plan to use data from automated meteorological stations operated by Mexico's Meteorological Service which are located at different elevations. However, there are not even

100 of these stations throughout Mexico, and their temporal coverage is limited (some stations were deployed in 2010, others in 2012 and so on).

Here are the minor and specific comments, line by line:

**Introduction:**

L30: "along the migratory route of the Monarch butterfly" Maybe this is too specific.

My idea is to show that climate data is used in studies that are not related to water resources and the Monarch butterfly is probably the most well known species of butterfly in North America.

L44: Terraclimate is regularly updated and now it is available until 2024.

Thanks for pointing this out; I have modified it.

L66: For CONUS, I think that PRISM deserves to be mentioned since it was one of the first and still one of the more reliable gridded datasets (https://prism.oregonstate.edu)

Thanks for your suggestion; I will include it.

**Methods:**

L101-102: how many stations was the final number?

The current version of the manuscript reads (L101-103) "… and once the climate records were in PostgreSQL only those stations with more than 10 years of registered data were selected; accordingly, not all available stations were used, and the number of stations varied across the 1951‑2021 period, as shown in Fig. 2."

Also, the caption of Fig. 2 reads "Number of weather stations used for daily interpolations of (a) temperature, (b) precipitation".

L108: regarding the outliers, wasn't any additional quality control performed? There's a lot of scientific literature on this.

For Mexico, the L15 dataset used those stations with > 50 days of data.

L108-109: how many stations were discarded?

The total number of stations with data is 5467; I will add this to the manuscript.

L110: As mentioned before about the use of KED independently for each day, it can generate further problems in long-term trends and temporal aggregations. Also, why 30 nearest stations and a 140km radius? The number of observations can greatly vary under these conditions. Lastly, was the internal coherence of temperature (TMAX>TMIN) checked after the interpolation, for each day? The above problems are especially important if a proper quality control was not performed to control the spatial coherence of the data (and I read nothing in this regard).

Regarding the selection of 30 stations and 180 km to apply $KED_l$, I will add the following lines:

"These values were recommended by Carrera-Hernandez et al., (2024) after a detailed comparison of different Kriging variants at both national and stratified domains showed that $KED_l$ using elevation as a secondary variable provided the best representation of yearly precipitation in Mexico." The manuscript that details these analyses is currently under review (Hydrological Sciences Journal).

The internal coherence of Tmax>Tmin was verified for each station before undertaking the interpolation. I will include the Tmax>Tmin coherence of the interpolated datasets on the revised manuscript.

L120-121: Is the code publicly available?

The code is not publicly available, but I can add an example to the new version of the manuscript.

**Validation:**

I expected a more complete validation since here only daily data considering the whole series was checked. For example, how the interpolation worked at different elevation ranges? or in different months? Did the method correctly predict the number of dry/wet days? Are monthly (or other) averages and standard deviation fit between predicted and observed values? These are the basic checks for any gridded dataset.

On the revised manuscript I plan to use independent data from automated meteorological stations located at different elevations. However, there are not even 100 of these stations throughout Mexico, and their temporal coverage is limited (some stations were deployed in 2010, others in 2012 and so on).

Figure 5: I am not sure how to interpret these graphics since, for example, $R^2$ needs complete series of predictions and observations to be compared but here you have one value per day/month/year

Indeed, Figure 5 shows the daily values of the coefficient of determination ($R^2$), the Coefficient of Efficiency (COE) and the Index of Agreement (IOA). These values were obtained through daily leave-one-out cross-validation. I will ad a flow diagram to show how the leave-one-out cross-validation was performed, along with how the semivariogram was modelled daily to perform the daily interpolations.

L242: why not comparing monthly or annual aggregates? or even trends? that would be more useful than comparing extreme events, which are not common (by definition) and the users may need a more regular use of the dataset.

I decided to use these extreme events because these events are important to study areas where flash floods occur and because floods (in general) are natural disasters in Mexico that need to be analyzed, In addition, these extreme events also cause landslides in Mexico´s mountainous regions.

L274-277: This comparison is not fair since you're comparing predictions with observations but only in the case of your dataset you know that the observation does not participate in the interpolation, but not in the rest of datasets. Furthermore, not all of them were built with the same observations, so it is hard to justify better results on your dataset.

I do not agree with this point; DAymet and L15 used some of the observations that I used to develop the MexHiResClimDB. In fact, I explain this issue on lines 277-281:

These performance statistics are shown in Fig. 7 along with their respective scattergrams of differences; however, it should be kept in mind that the performance metrics shown in the aforementioned figure for the MexHiResClimDB can not be directly compared with the metrics of Daymet, L15 or CHIRPS, because for the latter three cases some of the weather stations used to compute the metrics were used to develop the datasets – in summary, the performance metrics obtained through cross-validation are expected to be lower.

L298: what this function does?

I will describe what this function does in the revised manuscript. The r.univar command computes univariate statistics from the non-null cells of a raster map; these statistics are number of cells, minimum and maximum cell values, range, arithmetic mean, variance, standard deviation, coefficient of variation and sum.

L300: what was the threshold for considering a dry day? 0 mm / 0.1 mm / 0.001 mm?

Accoding to L300: "With this procedure, the ten wettest and driest days, months and years were obtained and summarized in Table 2 … ". The values of Table 2 are in $1\times10^9$ m$^3$ per day, month and year. I wanted to keep three decimals for all the columns and that is why the minimum daily values of precipitation shown in Table 2 do not change much. These values

are ordered in ascending order, and they were obtained by creating a table using the values obtained with the `r.univar` command.

L309: I dont see tendency in that table

I agree and in fact that is what can be read on line 309: ".. that no wettness or dryness tendency is easily seen on the values shown in Table 2"

L326-327: again, this is not a validation, just a comparison with other datasets that were not constructed with the same procedure. The only validation must be with the observations.

These lines refer to the performance metrics obtained with leave-one-out cross-validation that are shown on Fig. 8.

Lines 325-327 state the following: "To validate the interpolated temperature maps, the maximum and minimum values of Tmax (1998-6-15, 1967-1-10) and Tmin (2020-8-31, 1962-1-12) were selected to report their validation in detail. The results obtained with the leave-one-out crossvalidation are shown on Fig. 8, …"

L332-335: a visual comparison does not guarantee a correct validation.

I agree and will rephrase these lines accordingly.

L347: Fig 10 shows absolute values, but this is not trends. If you want to show trends you should calculate some statistics (Mann Kendall, Sen's slope) with their corresponding reliability value (p-value).

L350-351: this is not an acceptable way to indicate that there is a trend. Without statistical validation, this complete section must be removed.

I have modified Figure 10 and added a five-year moving average of the monthly and yearly anomalies in order to improve what is written on lines 347-351. The modified figure is shown as a separate file.

---

## Author Response (AR1)

Reply to reviews for manuscript ESSD-2025-100

**General comments**

I thank both reviewers for their thorough reviews. Both reviewers mentioned that the validation and comparison of the MexHiResClimDB had to be improved.

In particular, R1 mentioned that:

"The data and methods section needs substantial revision. Clear explanations, potentially a flowchart, and a discussion of how each step in the process might influence the results would be highly beneficial. The validation and comparison presented are limited and only applied to specific extreme cases. The manuscript lacks discussion of potential limitations of the datasets. It is not sufficiently demonstrated that the newly generated datasets are better than existing alternatives, despite multiple claims by the author, though we are confident this can be done with more convincing analyses

- Validation and comparison: The intercomparison with other datasets wasn't convincing. The author needs to make a more comprehensive comparison of their new dataset to other datasets, not only for selected extreme events. Further, they have to present how their data represents climate differently than other datasets and why it is so. I also suggest that the author discusses their methodological and data limitations.".

On the other hand, R2 mentions:

"I expected a more complete validation since here only daily data considering the whole series was checked. For example, how the interpolation worked at different elevation ranges? or in different months? Did the method correctly predict the number of dry/wet days? Are monthly (or other) averages and standard deviation fit between predicted and observed values? These are the basic checks for any gridded dataset.

- why not comparing monthly or annual aggregates? or even trends? that would be more useful than comparing extreme events, which are not common (by definition) and the users may need a more regular use of the dataset."

**Reply:**

I agree with both reviewers and accordingly, I used independent data from different Automatic Weather Stations (AWSs) located at different elevations and regions in Mexico. This new comparison appears in the following sections of the revised version of the manuscript (lines 350-434):

5.5.2 Validation with independent data from Automatic Weather Stations.

This subsection includes a figure showing both the location and elevation of the AWSs; it also includes a new Figure (Figure 10(a) and (b)), which shows the comparison of the precipitation mass curves for September 2013 and July 2010 obtained at different AWSs with those from the different gridded datasets compared in the manuscript. This section also has another new figure

(Fig. 11) which shows the hex-bin scatterplots that summarize the differences of daily precipitation registered at the AWSs and the precipitation sampled at their location for the gridded datasets compared in the manuscript. This section uses data from AWSs located near the Mexico-U.S.A. border to see how good are the data provided by the MexHiResClimDB in these areas, as only data from Mexico was used for its development (whereas the other datasets used data from weather stations located in the U.S.A.)

5.5.3 Monthly precipitation values at national and watershed levels.

This subsection shows the monthly accumulated precipitation at both the national and watershed level for September 2013 and July 2010. The two new figures of this subsection show the interpolation artifacts present in both L15 and Daymet, in particular for September 2013.

5.6 Validation and comparison of temperature.

This section was improved by also using temperature data from the AWSs and adding two figures: Figure 15 (a) and (b) shows the variation of daily Tmin and Tmax for September 2013 and July 2010 obtained with data from the AWSs and the datasets that provide temperature. The daily differences of Tmin and Tmax between data from AWSs and the gridded datasets are summarized in Fig. 16 (which is also a new figure).

I added a flowchart in order to describe how the daily interpolations were developed and how the results of cross-validation were stored in order to show the daily variation of the performance indices that appear in Fig. 6 (Fig. 5 in the previous version) in order to address the concern of R2, who mentioned in the review that "I am not sure how to interpret these graphics since, for example, $R^2$ needs complete series of predictions and observations to be compared but here you have one value per day/month/year"

Other general comments:

Regarding QC of the raw data, the following lines were added to the revised manuscript (lines 123-132):

> Although the removal of outliers considering neighbouring stations could have been done, this was not done due to the fact that precipitation in Mexico is highly variable within short distances due to the presence of hurricanes and these precipitation events need to be included in the gridded product. In addition, the final data selected by the previously mentioned procedure were not analysed for homogeneity and the station records were used without filling data gaps (i.e. data series reconstruction). Although gap-filling can be used to generate a complete data series for the considered time period – and thus keeping a uniform number of stations for the interpolation – it was decided to avoid it in order to use the original data, because the reconstruction process is generally based on weighted averages or modeling that consists of creating a reference

series formed as a weighting model of the data observed at neighbouring stations (Serrano-Notivoli and Tejedor, 2021), which is some type of interpolation (Daly, 2006). Further work can be done to address these issues and interested readers are referred to Serrano-Notivoli and Tejedor (2021) for a detailed analysis of QC on the development of gridded climate datasets.

Also, the following lines were added regarding the validation of gridded datasets (lines 163-168):

> In order to estimate the errors in spatial climate datasets, a combination of approaches should be used, involving data that are as independent from those used to generate the datasets along with common sense in the interpretation of results (Daly, 2006). Accordingly, the MexHiResClimDB was validated using: 1) leave-one-out cross validation – from which different performance metrics were obtained, 2) visual comparison of extreme precipitation events, 3) use of independent data acquired at different Automatic Weather Stations (AWSs) located throughout Mexico, and 4) comparison of the spatial distribution of accumulated monthly precipitation at both national and watershed scales.

Please note that the lines referred to by the reviewers correspond to the line numbers of the first manuscript.

**Reply to R1:**

General comments: The author presents a newly developed gridded, high-resolution climate dataset comprised of daily, monthly and yearly precipitation and temperature for Mexico that they have developed using stations data and Kriging with External Drift on a local neighborhood (KEDl) interpolation. The study presents a new dataset that can be very useful in understanding different aspects of climate change and its impacts at relatively high spatial and temporal resolution across Mexico.

Our main concerns relate to the filtering of the original datasets—specifically, the criteria used, the quality of the input data, and the details of the interpolation method. The data and methods section needs substantial revision. Clear explanations, potentially a flowchart, and a discussion of how each step in the process might influence the results would be highly beneficial. The validation and comparison presented are limited and only applied to specific extreme cases. The manuscript lacks discussion of potential limitations of the datasets. It is not sufficiently demonstrated that the newly generated datasets are better than existing alternatives, despite multiple claims by the author, though we are confident this can be done with more convincing analyses.

As mentioned on the first page of this reply, I improved the validation by using independent data from different Automatic Weather Stations (AWSs) located at different elevations and regions in Mexico. This new comparison appears on lines 350-434.

Validation and comparison: The intercomparison with other datasets wasn't convincing. The author needs to make a more comprehensive comparison of their new dataset to other datasets, not only for selected extreme events. Further, they have to present how their data represents climate differently than other datasets and why it is so. I also suggest that the author discusses their methodological and data limitations.

To address this concern, I added the following subsections:

5.5.2 Validation with independent data from Automatic Weather Stations.

This subsection includes a figure showing both the location and elevation of the AWSs; it also includes a new Figure (Figure 10(a) and (b)), which shows the comparison of the precipitation mass curves for September 2013 and July 2010 obtained at different AWSs with those from the different gridded datasets compared in the manuscript. This section also has another new figure (Fig. 11) which shows the hex-bin scatterplots that summarize the differences of daily precipitation registered at the AWSs and the precipitation sampled at their location for the gridded datasets compared in the manuscript. This section uses data from AWSs located near the Mexico-U.S.A. border to see how good are the data provided by the MexHiResClimDB in these areas, as only data from Mexico was used for its development (whereas the other datasets used data from weather stations located in the U.S.A.)

5.5.3 Monthly precipitation values at national and watershed levels.

This subsection shows the monthly accumulated precipitation at both the national and watershed level for September 2013 and July 2010. The two new figures of this subsection show the interpolation artifacts present in both L15 and Daymet, in particular for September 2013.

5.6 Validation and comparison of temperature.

This section was improved by also using temperature data from the AWSs and adding two figures: Figure 15 (a) and (b) shows the variation of daily Tmin and Tmax for September 2013 and July 2010 obtained with data from the AWSs and the datasets that provide temperature. The daily differences of Tmin and Tmax between data from AWSs and the gridded datasets are summarized in Fig. 16 (which is also a new figure).

Other comments:

In the comments below. L indicate line number, * indicate major comments.

- L3: remove terms like largest , highest etc and provide specific values like what temporal coverage, especially in abstract.

I agree and will remove these adjectives where possible.

L10-17: not clear if the author is just presenting these values or saying that these values are more realistic than values from other datasets. I am not clear what the author is trying to say by saying 'a summary that was not available before'. It is available from other datasets like ERA5land. Reliability of the values can be different, but it is available.

I am presenting these values because to date there is no information anywhere regarding country-wide information on climate extremes for Mexico's. The information could be extracted from ERA5land, but to my knowledge no one has summarized this info before.

L12-17: for which period are these values true, 1951-2020? specify.

Yes, those values are for the 1951-2020 period; I did not want to mention again "for the 1951-2020 period…", because I thought it would be too repetitive.

- L46-48: redundant
- L46 -48: superfluous. This whole paragraph can be summarized to few sentences, much of the details may not be relevant to the paper

When I first read this suggestion, I agreed with it, but when I started working on the revised version of the paper, I thought that the introduction would have a large table that would make for an arid read and decided against it. I added the following lines to this section:

"For a more in-depth review of gridded climate products, interested readers are referred to the recent work of Mankin et al. (2025), who reviewed a total of 63 gridded climate datasets"

- *L63 (whole paragraph): I suggest that the author create a table with name of dataset, resolution ( spatial , temporal), data period, region and citation . such table will give this exact information from this and previous paragraphs and will be more succinct

I have the same reply as for the previous point.

- L75: "monthly surfaces of precipitation, Tmin and Tmax for the 1910–2009 period (i.e. 12 surfaces in total)". If these were normals, wouldn't this be 36 surfaces? But also, the term "monthly surfaces" is ambiguous about whether it is a monthly gridded time series or climatological average/normal. As suggested, and table would be more succinct and explicit.

I rephrased these lines to "developed monthly surfaces of precipitation, Tmin and Tmax for 1910‑2009 (i.e. 12 surfaces in total)"

- *L80: what about Daymet / ERA5 or other regional and  global datasets that covers Mexico and available at daily resolution  but may not be as same spatial resolution as this dataset ? However,  What problem their spatial resolution creates that this

MexHiresClimDB will better address ? Need to explain why this work is important and what value it adds in more detail.

This need is stated on lines 85-88: "From this summary, it can be seen than to date, there is not a daily high resolution climate dataset available for Mexico, which is why the Mexico's High Resolution Climate Database (MexHiresClimDB), which covers the 1951‑2020 period at a spatial resolution of 20'' (≈600) meters was developed"

The issue is also addressed on Figures 12 and 13, which show the advantage of having a climate dataset with higher resolution that can adequately represent the spatial variability of climate variables.

• L93: use numeric for XXth

Thank you for noticing this; I have already modified it.

• L95: what is the link between the description of study area and the datasets the author developed ?

Thank you for pointing this out; I have already modified the manuscript and it now reads as follows:

Due to Mexico's geographic context, its precipitation is extremely variable, as on a given day during the rainy season, precipitation can vary from 0 to more than 300 millimetres and the adequate representation of these extreme events is important in Mexico, in particular for those cities that depend on reservoirs that are only filled during hurricanes or where flash-floods are a recurrent event.

• L100: what aforementioned refer to ? Provide the name of dataset. Throughout the paper try to use 'aforementioned' less often as it creates confusion

Thanks for the suggestion; this "aforementioned" refers to the database that I developed in PostgreSQL. These lines will be improved with the flow chart that I will include in the new version of the manuscript.

• L100-101 : rewrite: how many stations originally were there , how many were used ? what are the criteria to select the stations

Lines 111-117 now read:

once the climate records were in PostgreSQL only those stations with more than 80% of registered data for at least 10 continuous years were selected (wheras the L15 dataset (Livneh et al., 2015) used a 50-day threshold for stations in Mexico). Accordingly, the number of stations used to undertake the interpolations across the 1951‑2021 period varies, as shown in

Fig. 3. This Figure shows that there are fewer records of temperature than precipitation (a situation also observed for North America by Tang et al. (2020)) and that the maximum number of records for both variables were registered in years 1982 and 1983 (with over 4000 daily records for precipitation and between 3000‑3500 for temperature out of a total of 5467 stations from the downloaded raw data), and that years 1951 and 2021 exhibit the lowest number of records

Also, the caption of Fig. 2 reads "Number of weather stations used for daily interpolations of (a) temperature, (b) precipitation".

- L108: what might have caused less number of stations after 2021 ?

I think that this was caused by a reduction in budget and the believe that in situ measurements are not needed.

- *Figure 2. The station coverage in this dataset is much better than in the CRUts and GPCC datasets, which report a dramatic (>90%) decline in station coverage after 1998. This is a compelling advantage of the MexiHiRes product and it would be useful to highlight it.

Thank you for your suggestion.

*L110 onwards to paragraph: need more description of the process and how this method works. For example, what happens to temperature lapse rate and what value is used ? A flowchart with all the steps involved from data collection to final output will be useful.

Kriging with External Drift considers the relationship between temperature and elevation, because elevation is used as an auxiliary variable.

*L125-128: can the author show how sensitive their used values are ?

The following was added to lines 147-155:

Based on the author's previous work on nationwide interpolation of yearly precipitation in Mexico, a cut-off distance of 180 km and a local neighbourhood of 30 stations was used, because KEDl with these parameters adequately interpolates precipitation even when this process is anisotropic (Carrera-Hernandez and Gaskin, 2007; Carrera-Hernandez et al., 2025). These parameters were recommended by Carrera-Hernandez et al. (2025) after a detailed comparison of different Kriging variants at the national level using both global and stratified domains that used different auxiliary variables with a combination of different cut-off distances and local neighbourhoods. This comparison showed that KEDl using elevation as a secondary variable with a cut-off distance of 180 km and a local neighbourhood of 30 stations provided the best representation of yearly precipitation in Mexico

- L129-133: can be more succinct

I think that it is important to show the amount of time and resources required to develop this database, but will try to rephrase it.

- L134: recommend re-expressing $1.227 \times 10^6$ as 28 months so readers don't have to do the math. It's a lot of computation!

Thanks for the suggestion; I agree with it.

- L129: Is 26GB minimum requirement? not clear

Each interpolation takes 26GB of RAM, but I will rephrase it.

- L146: "if R2=0.80, then the model explains 80% of the variability". Reword: it explains 80% of variance, not variability. since variance is calculated from squared deviations, this interpretation will overestimate the amount of variability/dispersion that is explained

Thanks for your suggestion; I will reword this line.

- L170: is it value or variable, be specific.

Thanks for this observation; it is variables and I have already corrected it.

- L172: it's true that MAE for raw precipitation (in mm) is meaningless, since an error of 50mm has a different significance in a wet vs dry climate. however, MAE is meaningful if precipitation is log-transformed prior to analysis of error. By the same token, I would expect that the other metrics (R-square, COE, and IOA) are confounded by raw precipitation values in mm; wouldn't they primarily represent wetter regions—and wet anomalies—where absolute errors (in mm) are larger.

The MAE is shown on the new validation section.

- *L200-239: I suggest the author move the description of these datasets to introduction and summarize there. Here they directly present the results of comparison.

Thanks for your suggestion; however, I tried it and did not like how it turned out.

- *L290-291: This is not well justified statement but based on selected extreme event and a single coefficient. How this statement compares with what they presented in figure 5. I recommend performing relative Root mean Square Error to explore relative goodness of each dataset for both temperature and precipitation

On lines 281-291 I compare both the COE and the IOA. I use the extreme events of Figure 6 because I believe that these extremes need to be well represented in gridded databases. I do not agree with R1's suggestion of using the RMSE in addition to the metrics shown in the manuscript (R2, COE and IOA), due to the reasons given on lines 160-165:

L195-198: The Mean Absolute Error (MAE) is also used in this work because it is an unambiguous and more natural measure of average error than the Root Mean Square Error (RMSE, Willmott and Matsuura (2005)) due to the bias of RMSE when large outliers are present (Legates and McCabe, 1999) and because the RMSE does not describe average error alone and its use has been discouraged (Willmott and Matsuura, 2005).

L200: The modified Index of Agreement (IOA, Legates and McCabe (1999)) has the advantage that errors and differences are not inflated by their squared values…

L203-204: Another advantage of the IOA is that it is related to the Mean Average Error (MAE) and the Mean Absolute Deviation (MAD)

- Fig 7. The panels are labelled a,b,c,b,c.

Thank you for pointing this error; I have corrected it.

- Fig 7. It is very hard to see the distributions at typical printing size. Recommend altering the color scheme to better stand out against the background.

Thanks for your suggestion; I modified the colors of all the scattergrams

L311: not clear what the author is trying to say here.

Thank you for pointing this out. I have corrected the manuscript. It now reads (l 437):

The MexHiResClimDB also includes temperature data, and the analysis of extreme values of temperature is useful in climate change analysis; however, there is currently no information available on the coldest or hottest days in Mexico for the 1951--2020 period.

- *L325-326: similar comments for precipitation and temperature. I suggest the author adds average comparison as well as relative comparison among the datasets to provide better perception on how their new dataset is better than existing datasets, if it is.

The validation and comparison section was improved by using independent data from the Automatic Weather Stations.

- L328: no need to write the full form again, be consistent

I decided to write the full form again to avoid monotony to the reader; I changed it to IOA in the new version of the manuscript.

- L330: "it is not possible to obtain cross-validation values for L15 or Daymet". This is a reason to use some other metric for intercomparison.

The validation is now improved with data from the Automatic Weather Stations.

- L335: "neither L15 nor Daymet were capable of howing the temperature extremes that were obtained through the MexHiResClimDB". This is not apparent from figure 9. The maximum values seem similar to L15

I removed Figure 9 and these lines do not appear in the new version of the manuscript.

- Figure 9: it is not clear what the author is trying to illustrate from these plots. It is already clear that their data has longer temporal coverage across Mexico than other mentioned datasets. It does not provide information on how better their dataset is compared to others.

I decided to remove this figure because I now use the data from the Automatic Weather Stations.

- Figure 11: describe what climate stripes bars represent and what additional information they provide other than the anomaly plots. Otherwise remove

I decided to also show the climate stripes in order to clarify where they come from. I do not agree with R1's suggestion of removing them.

- L393-L395: this has only been demonstrated for a few specific events, so the statement should likely be qualified as such.

I agree and modified the lines accordingly to the following (l 524-526):

… the daily precipitation data provided by this database is the only one that adequately represents the spatial variation of the extreme precipitation events that occurred during September 15--16 of 2013, caused by the presence of Tropical storm Manuel in the Pacific Ocean and Hurricane Ingrid (Cat 1) in the Gulf of Mexico.

**Reply to R2**

The author develops a complete daily gridded dataset of precipitation and temperature for Mexico at a very high resolution considering the extension of the spatial domain. The research is mostly correct; however, I have some concerns about the method used to develop the daily grid and the validation process.

I thank R2 for these positive comments and for the time taken to review this manuscript.

Here are the minor and specific comments, line by line:

**Introduction:**

L30: "along the migratory route of the Monarch butterfly" Maybe this is too specific.

My idea is to show that climate data is used in studies that are not related to water resources; however, I modified it and not it read "along migratory routs of butterflies…"

L44: Terraclimate is regularly updated and now it is available until 2024.

Thanks for pointing this out; I have modified it.

L66: For CONUS, I think that PRISM deserves to be mentioned since it was one of the first and still one of the more reliable gridded datasets (https://prism.oregonstate.edu)

Thanks for your suggestion; it appears on line 68-69

**Methods:**

The current version of the manuscript reads (L101-103) "… and once the climate records were in PostgreSQL only those stations with more than 80% of registered data for at least 10 continuous years were selected. Accordingly, the number of stations fused to undertake the interpolations across the 1951-2021 period varies, as shown in Fig. 3". Also, the caption of Fig. 2 reads "Number of weather stations used for daily interpolations of (a) temperature, (b) precipitation".

L108: regarding the outliers, wasn't any additional quality control performed? There's a lot of scientific literature on this.

L116 now reads: "A basic Quality Control (QC) was applied to the generated database, with daily precipitation values above 600 mm, as well as temperature values below -30 C or above 60 C discarded, along with those days where Tmin > Tmax."

As previously mentioned, lines 123-132 show the following:

> Although the removal of outliers considering neighbouring stations could have been done, this was not done due to the fact that precipitation in Mexico is highly variable within short distances due to the presence of hurricanes and these precipitation events need to be included in the gridded product. In addition, the final data selected by the previously mentioned procedure were not analysed for homogeneity and the station records were used without filling data gaps (i.e. data series reconstruction). Although gap-filling can be used to generate a complete data series for the considered time period – and thus keeping a uniform number of stations for the interpolation – it was decided to avoid it in order to use the original data, because the reconstruction process is generally based on weighted averages or modeling that consists of creating a reference series formed as a weighting model of the data observed at neighbouring stations (Serrano-Notivoli and Tejedor, 2021), which is some type of interpolation (Daly, 2006). Further work can be done to address these issues and interested readers are referred to Serrano-Notivoli and Tejedor (2021) for a detailed analysis of QC on the development of gridded climate datasets.

As mentioned on l. 116, the total number of stations was 5467, and the discarded number varies, as shown in Fig. 3.

L110: As mentioned before about the use of KED independently for each day, it can generate further problems in long-term trends and temporal aggregations. Also, why 30 nearest stations and a 140km radius? The number of observations can greatly vary under these conditions. Lastly, was the internal coherence of temperature (TMAX>TMIN) checked after the

interpolation, for each day? The above problems are especially important if a proper quality control was not performed to control the spatial coherence of the data (and I read nothing in this regard).

The following was added to lines 147-155:

> Based on the author's previous work on nationwide interpolation of yearly precipitation in Mexico, a cut-off distance of 180 km and a local neighbourhood of 30 stations was used, because KEDl with these parameters adequately interpolates precipitation even when this process is anisotropic (Carrera-Hernandez and Gaskin, 2007; Carrera-Hernandez et al., 2025). These parameters were recommended by Carrera-Hernandez et al. (2025) after a detailed comparison of different Kriging variants at the national level using both global and stratified domains that used different auxiliary variables with a combination of different cut-off distances and local neighbourhoods. This comparison showed that KEDl using elevation as a secondary variable with a cut-off distance of 180 km and a local neighbourhood of 30 stations provided the best representation of yearly precipitation in Mexico
>
> The temperature summary of Figure 17 show that Tmax > Tavg > Tmin

L120-121: Is the code publicily available?

I added a snippet, as shown in Section 8, Code Availability.

**Validation:**

I expected a more complete validation since here only daily data considering the whole series was checked. For example, how the interpolation worked at different elevation ranges? or in different months? Did the method correctly predict the number of dry/wet days? Are monthly (or other) averages and standard deviation fit between predicted and observed values? These are the basic checks for any gridded dataset.

I used independent data from different Automatic Weather Stations (AWSs) located at different elevations and regions in Mexico. This new comparison appears in lines 350-434 of the revised version of the manuscript.

Figure 5: I am not sure how to interpret these graphics since, for example, $R^2$ needs complete series of predictions and observations to be compared but here you have one value per day/month/year

Indeed, Figure 6 (previously Fig. 5) shows the daily values of the coefficient of determination ($R^2$), the Coefficient of Efficiency (COE) and the Index of Agreement (IOA). These values were obtained through daily leave-one-out cross-validation. The flowchart of Figure 2 tries to clarify this point.

L242: why not comparing monthly or annual aggregates? or even trends? that would be more useful than comparing extreme events, which are not common (by definition) and the users may need a more regular use of the dataset.

I decided to use these extreme events because these events are important to study areas where flash floods occur and because floods (in general) are natural disasters in Mexico that need to be analyzed. However, the new validation developed with data from the AWSs show the precipitation mass curves for two months.

L274-277: This comparison is not fair since you're comparing predictions with observations but only in the case of your dataset you know that the observation does not participate in the interpolation, but not in the rest of datasets. Furthermore, not all of them were built with the same observations, so it is hard to justify better results on your dataset.

I do not agree with this point; Daymet and L15 used some of the observations that I used to develop the MexHiResClimDB. In fact, I explain this issue on lines 314-319:

> These performance statistics are shown in Fig. 7 along with their respective scattergrams of differences; however, it should be kept in mind that the performance metrics shown in the aforementioned figure for the MexHiResClimDB can not be directly compared with the metrics of Daymet, L15 or CHIRPS, because for the latter three cases some of the weather stations used to compute the metrics were used to develop the datasets – thus, the performance metrics obtained through cross-validation are expected to be lower.

L298: what this function does?

The use of the r.univar command appears now in line 356, which reads "the r.univar command of the GRASS GIS – which computes univariate statistics (such as minimum, maximum or total sum) from the non-null cells of a raster map"

L300: what was the threshold for considering a dry day? 0 mm / 0.1 mm / 0.001 mm?

Accoding to L339: "With this procedure, the ten wettest and driest days, months and years were obtained and summarized in Table 2 … ". The values of Table 2 are in $1\times10^9$ m$^3$ per day, month and year. I wanted to keep three decimals for all the columns and that is why the minimum daily values of precipitation shown in Table 2 do not change much. These values are ordered in ascending order, and they were obtained by creating a table using the values obtained with the `r.univar` command.

L309: I dont see tendency in that table

I agree and in fact that is what can be read on line 346: ".. that no wettness or dryness tendency is easily seen on the values shown in Table 2"

L326-327: again, this is not a validation, just a comparison with other datasets that were not constructed with the same procedure. The only validation must be with the observations.

These lines are now lines 448-450 and they refer to the performance metrics obtained with leave-one-out cross-validation that are shown on Fig. 14

Lines 448-451 state the following: "To validate the interpolated temperature maps, the maximum and minimum values of Tmax (1998-6-15, 1967-1-10) and Tmin (2020-8-31, 1962-1-12) were selected to report their validation in detail. The results obtained with the leave-one-out cross-validation are shown on Fig. 14, …"

L332-335: a visual comparison does not guarantee a correct validation.

I agree; I removed that paragraph because the validation is now done with data from the AWSs

L347: Fig 10 shows absolute values, but this is not trends. If you want to show trends you should calculate some statistics (Mann Kendall, Sen's slope) with their corresponding reliability value (p-value).

This is now Figure 17 and I added the 1961-1990 normals to the monthly plots, as well as the monthly anomalies.

L350-351: this is not an acceptable way to indicate that there is a trend. Without statistical validation, this complete section must be removed.

These are now lines 481-482 and I rephrased them to "However, the bottom of this hourglass (for the three temperatures reported) is narrower than it is at its top, which clearly shows an increase in temperature".

---

## Author Response (AR2)

Reply to reviews for Manuscript ESSD-2025-100-R1

General comments:

I thank both reviewers for taking the time to review the revised version of the manuscript. R1 recommended accepting the manuscript, while R3 provided minor comments that improved the manuscript. Following the comments provided by R3, I changed the size of the text on Figs. 5 and 6 to improve their readability and included all the suggested style corrections.

Finally, R3 raised the following questions:

*Line 206 -212 Why are there systematic differences in the MAE values? Does the difference in MAE values reflect the variability in the magnitude of the temperature values or the variability over time in the temperature values during certain seasons?*

*Lines 213 – 220 Is a higher value performance indicator a reflection of more stable weather or better measurement accuracy? I.e. is there a reason for the systematic difference?*

In order to address these questions, I added the following to lines 213-215:

The fact that the MAE exhibits lower values in summer for both Tmin and Tmax reflect a more "stable" spatial variation of temperature for that season; this occurs because during autumn and winter Mexico experiences cold fronts, which can not be related to elevation and affect only parts of the country

And the following to lines 223-225:

Heterogeneity of the climate variables affect the computed performance indices, and a more dense coverage improve their estimation, as indicated by better performance values between 1975‑1985.